



# Accurate simulation of transient landscape evolution by eliminating numerical diffusion: the TTLEM 1.0 model

Benjamin Campforts[1], Wolfgang Schwanghart[2] and Gerard Govers[1]

[1] KU Leuven, Division Geography, Department of Earth and Environmental Sciences

[2] Universität Potsdam, Institute for Earth and Environmental Science

*Correspondence to*: benjamin.campforts@kuleuven.be

**Abstract**. Landscape evolution models (LEM) allow studying the earth surface response to a changing climatic and tectonic forcing. While much effort has been devoted to the development of LEMs that simulate a wide range of processes, the

numerical accuracy of these models has received much less attention. Most LEMs use first order accurate numerical methods that suffer from substantial numerical diffusion. Numerical diffusion particularly affects the solution of the advection equation and thus the simulation of retreating landforms such as cliffs and river knickpoints with potential unquantified consequences for the integrated response of the simulated landscape. Here we present TTLEM, a spatially explicit, raster based LEM for the study of fluvially eroding landscapes in TopoToolbox 2. TTLEM prevents numerical diffusion by implementing a higher order

flux limiting total volume method that is total variation diminishing (TVD-TVM) and solves the partial differential equations of river incision and tectonic displacement. We show that the choice of the TVD-TVM to simulate river incision significantly influences the evolution of simulated landscapes and the spatial and temporal variability of catchment wide erosion rates. Furthermore, a 2D TVD-TVM accurately simulates the evolution of landscapes affected by lateral tectonic displacement, a process whose simulation is hitherto largely limited to LEMs with flexible spatial discretization. By providing accurate

numerical schemes on rectangular grids, TTLEM is a widely accessible LEM that is compatible with GIS analysis functions from the TopoToolbox interface.

## 1. Introduction

Landscape evolution models (LEMs) simulate how the earth surface evolves in response to different driving forces including

tectonics, climatic variability and human activity. LEMs are integrative as they amalgamate empirical data and conceptual models into a set of mathematical equations that can be used to reconstruct or predict terrestrial landscape evolution and corresponding sediment fluxes (Howard, 1994). Studies that address how climate variability and land use changes will affect landscapes on the long term increasingly rely on LEMs (Gasparini and Whipple, 2014).

A large number of geophysical processes act on the earth surface, mostly driven by gravity and modulated by the presence of

water, ice and organisms (Braun and Willett, 2013). These processes critically depend on the availability potential energy, brought into or withdrawn from the landscape by tectonic forces (Wang et al., 2014). Weathering and erosion respond to tectonic uplift, shaping the landscape through the lateral transport of sediments and, to a certain degree, also through feedback on regional uplift patterns (Whipple and Meade, 2004).

LEMs allow to integrate growing field evidence covering different spatial and temporal timescales (Glotzbach, 2015), thereby

accommodating a broad range of applications with fundamental importance in the development of geosciences (Bishop, 2007). LEMs are key to understanding landscape evolution both over time scales of millions of years (van der Beek and Braun, 1998;



Tucker and Slingerland, 1994; Willett et al., 2014; Willgoose et al., 1991b) and much shorter, millennial, timescales (Coulthard et al., 2012). LEMs simulate the interaction between different processes and provide insights into how these interactions result in different landforms. Moreover, visualizing LEM output in intuitive animations stimulates the development of new theories
and hypotheses (Tucker and Hancock, 2010). LEMs have also successfully been used for higher education in geomorphology and geology, improving students understanding of geophysical processes (Luo et al., 2016).

Landscape evolution is not always smooth and gradual. Instead, sudden tectonic displacements along tectonic faults can create distinct landforms with sharp geometries (Whittaker et al., 2007). These topographic discontinuities are not necessarily smoothed out over time, but may persist over long time scales in transient landscapes (Mudd, 2016). For example, faults may
spawn knickpoints along river profiles. These knickpoints will propagate upstream as rapids or water falls (Hoke et al., 2007), thereby maintaining their geometry through time (Campforts and Govers, 2015). After an uplift pulse, the river will only regain a steady state when the knickpoint finally arrives in the uppermost river reaches. Transiency is not limited to individual rivers but also affects larger systems such as the Southern Alps of New Zealand where the landscape may never reach a condition of steady state due to the permanent asymmetry in vertical uplift, climatically driven denudation and horizontal tectonic advection
(Herman and Braun, 2006).

Topographic discontinuities that result from transient 'shocks' are inherently difficult to model accurately. Most of the widely applied LEMs (Valters, 2016), use first order accurate explicit or implicit finite difference methods to solve the partial differential equations (PDE) that are used to simulate river incision. These schemes suffer from numerical diffusion (Campforts and Govers, 2015; Royden and Perron, 2013). Numerical diffusion will inevitably lead to the gradual disappearance of
knickpoints: the inherent inaccuracy of (implicit) first order accurate methods will result in ever smoother shapes. While this topographic smearing has already been shown to have implications for the accuracy of modelled longitudinal river profiles, we hypothesize that it is also relevant for the simulation of hillslope processes: hillslopes respond to river incision and, thus, inaccuracies in river incision modelling will propagate to the hillslope domain. Whether and to what extent this occurs, is yet unexplored.

Tectonic displacement is similar to river knickpoint propagation; in both cases, sharp landscape forms are laterally moving. Numerical diffusion may therefore significantly alter landscape features when tectonic shortening or extension if simulated using first order accurate methods. This problem can in principle be overcome with flexible gridding, whereby the density of nodes on the modelling domain is dynamically adapted to the local rate of change in topography. However, models using flexible gridding have other constraints. They are much more complex to implement and hence less easy to adapt, require
permanent mesh grid updates and impose the structure of the numerical grid to the natural drainage network as rivers are forced to follow the numerically composed grid structure. Furthermore, the output of flexible grid models is not directly compatible for most software that is available for topographic analysis (Schwanghart and Kuhn, 2010).

Here we present TTLEM, a spatially explicit raster based LEM, which is based on the object-oriented function library TopoToolbox 2 (Schwanghart and Scherler, 2014). Contrary to previously published LEMs we solve river incision using a
flux limiting total volume method (TVM) which is total variation diminishing (TVD) in order to prevent numerical diffusion when solving the stream power law. Our numerical scheme expands on previous work (Campforts and Govers, 2015) by extending the mathematical formulation of the TVD method from 1D to entire river networks. Moreover, we developed a 2D TVD-FVM to simulate horizontal tectonic displacement on regularly grids, thus allowing accounting for three dimensional variations in tectonic deformation. The objective of this paper is to evaluate TTLEM and assess the performance of the
numerical methods to a variety of real-world and synthetic situations. We show that the use of this updated numerical method has implications for the simulation of both catchment wide erosion rates and landscape topography over geological time scales.





TTLEM provides the geoscientific community with an easily accessible and adaptable tool. TTLEM is therefore a fully open source software package, written in MATLAB and based on the TopoToolbox platform. Users should be able to run TTLEM using both real data and synthetic landscapes. Moreover, the integration of TTLEM in TopoToolbox allows direct digital terrain analysis using the TopoToolbox library (Schwanghart and Scherler, 2014). In its current form TTLEM is limited to uplifting, fluvially eroding landscapes: further development will allow to integrate other processes (e.g. glacial erosion) as well as the explicit routing of sediment through the landscape.

## 2. Theory and geomorphic transport laws

### 2.1. Tectonic deformation

In its most simplest form, tectonic deformation is represented by vertical uplift, $U(x,y,t)$ [L t$^{-1}$]. However, many tectonic configurations imply that displacements have both a vertical (uplift or subsidence) and a lateral (extension or shortening) component (Willett, 1999; Willett et al., 2001). The change in elevation of the earth surface ($z$) over time due to tectonic deformation is then:

$$\frac{\partial z}{\partial t} = U + \mathbf{v_x} \frac{\partial z}{\partial x} + \mathbf{v_y} \frac{\partial z}{\partial y} \tag{1}$$

where $v_x$ and $v_y$ [L T$^{-1}$] are the tectonic displacement velocities in the $x$ and $y$ direction, respectively.

### 2.2. River incision

Detachment limited fluvial erosion is calculated based on the well-established relation between the channel gradient and the contributing drainage area ($A$), also referred to as the Stream Power Law (SPL) (Howard and Kerby, 1983):

$$\frac{\partial z}{\partial t} = -K w_K \left( w_A A \right)^m \left( \frac{\partial z}{\partial x} \right)^n \tag{2}$$

$K$ [L$^{1-2m}$ t$^{-1}$] is an erodibility parameter that depends on local climate, hydraulic roughness, lithology and sediment load. $K$ can be adapted to local variations in erodibility by using a scaling coefficient $w_K$ [dimensionless]. In case of uniform erodibility, $w_K$ is set to one. $A$ is the drainage area, which is used as a proxy for the local discharge. Similar to $K$, $A$ can be corrected for regional precipitation variabilities through a scaling coefficient $w_A$ [dimensionless]. $m$ and $n$ represent the area and slope exponent: their values reflect hydrological conditions, channel width, as well as the dominant erosion mechanism. $K$, $m$ and $n$ are interdependent and it is usually impractical to constrain any of their values alone (Croissant and Braun, 2014; Lague, 2014). Thus, many studies provide estimates for the $m/n$ ratio. For $m/n$ ratios between 0.35 and 0.8, $K$ values span several orders of magnitude between $10^{-10}$ - $10^{-3}$ m$^{(1-2m)}$ yr$^{-1}$ (Kirby and Whipple, 2001; Seidl and Dietrich, 1992; Stock and Montgomery, 1999). In order to represent fluvial sediment transport, it has previously been proposed to add a diffusion component (Rosenbloom and Anderson, 1994). However, we follow others in assuming that in eroding settings, detachment limited erosion is controlling landscape evolution and is represented by the advection equation represented in Eq. (2) (Attal et al., 2008; Goren et al., 2014; Howard and Kerby, 1983; Whipple and Tucker, 1999).

### 2.3. Hillslope processes

River incision drives the development of erosional landscapes by changing the base level for hillslope processes. Steepening of hillslopes subsequently leads to increased sediment fluxes from hillslopes to the river system. Hillslope erosion is equal to the divergence of the flux of soil/regolith material ($\mathbf{q_s}$, [L$^3$ L$^{-1}$ T$^{-1}$]):

$$\frac{\partial z}{\partial t} = -\nabla \mathbf{q_s} \tag{3}$$





Different geomorphological laws describe hillslope response to lowering base levels. The model of linear diffusion assumes that the soil/regolith flux is proportional to the hillslope gradient (Culling, 1963):

$$\mathbf{q_s} = -D\nabla z \qquad (4)$$

where $D$ is the diffusivity $[L^2 t^{-1}]$ that parameterizes hillslope erosivity and erodibility and determines rate of soil/regolith creep. Linear hillslope diffusion produces convex upward slopes. Field evidence, however, suggests that this model is only rarely

appropriate (Dietrich et al., 2003). Instead, hillslopes often tend to have convex-planar profiles because rapid, ballistic particle transport and shallow landsliding dominate as soon as slopes approach or exceed a critical angle (DiBiase et al., 2010; Larsen and Montgomery, 2012). To account for this rapid increase of flux rates with increasing slopes, Andrews and Bucknam (1987) and Roering et al. (1999) proposed a nonlinear formulation of diffusive hillslope transport, assuming that flux rates increase to infinity if slope values approach a critical slope $S_c$:

$$\mathbf{q_s} = -\frac{D\nabla z}{1 - \left(|\nabla z|/S_c\right)^2} \qquad (5)$$


Main controls on variations of $D$ include substrate, lithology, soil depth, climate and biological activity, amongst others. Values of $D$ vary widely and range between $10^{-3}$ and $10^{-1}$ m$^2$ yr$^{-1}$ for slopes under natural land use (Campforts et al., 2016; DiBiase and Whipple, 2011; Jungers et al., 2009; Roering et al., 1999; West et al., 2013).

### 2.4. Overall landscape evolution

In summary, TTLEM solves the following partial differential equations: First, it simulates the horizontal tectonic displacements over the entire model domain:

$$\frac{\partial z}{\partial t} = \mathbf{v_x}\frac{\partial z}{\partial x} + \mathbf{v_y}\frac{\partial z}{\partial y} \qquad (6)$$

Second, TTLEM simulates detachment limited river incision for the parts of the landscape that are predominantly sculpted by fluvial processes. We determine that domain where contributing drainage area ($A$) exceeds a critical drainage area ($A_c$):

$$\frac{\partial z}{\partial t} = U + \left(v_x\frac{\partial z}{\partial x} + v_y\frac{\partial z}{\partial y}\right) - \left(Kw_K\left(w_A A\right)^{(m+\mathrm{var}(m))}\left(\frac{\partial z}{\partial x}\right)^n\right) \qquad (7)$$


where var($m$) refers to the variability on $m$ which is explained further (Eq. (20) ).

Third, we define the hillslope domain where $A < A_c$. Topographic changes in this domain are calculated by:

$$\frac{\partial z}{\partial t} = \frac{\rho_r}{\rho_s}U - \nabla\mathbf{q_s} \qquad (8)$$





where $\rho_r$ and $\rho_s$ are the bulk densities of the bedrock and the regolith material, respectively [m L$^3$]. The formulation of Eq. (8) implies that we assume that hillslopes are generally covered by regolith and/or soil.


### 3. Implementation and numerical schemes of TTLEM

Our main motivation to develop TTLEM is to provide users with a multi-process landscape evolution model that has a good overall computational performance and high numerical accuracy. TTLEM is predominantly written in the MATLAB programming language; to reduce run times, however, TTLEM encompasses some C-code where this significantly improves
performance. Integrating TTLEM into TopoToolbox enables running the model, visualizing and analyzing its output in the same computational environment.

Figure 1 shows a schematic representation of the TTLEM workflow. Users can configure the tectonic setting by providing (i) a 2D or 3D array that represents spatially and spatio-temporally variable vertical uplift patterns, respectively, and (ii) two matrices to represent horizontal velocity fields ($v_x$ and $v_y$). TTLEM accepts synthetic topographies and real world DEMs and
leaves users with full control on model parameter values. In the following sections, we will discuss the numerical methods involved in TTLEM to solve the PDEs described in section 2. The section numbers correspond to the processes indicated in the workflow in Fig. 1.

### 3.1. Drainage network development

TopoToolbox provides a function library for deriving and updating the drainage network and terrain attributes in MATLAB
(Schwanghart and Scherler, 2014). The calculation of flow-related terrain attributes, i.e., data derived from flow directions, relies on a set of highly efficient algorithms that exploit the directed and acyclic graph structure of the river flow network (Phillips et al., 2015). Nodes of the network represent grid cells and edges represent the directed flow connections between the cells in downstream direction. Topological sorting of this network of grid cells transforms returns an ordered list of cells in that upstream cells appear before their downstream neighbors. Based on this list, we calculate terrain attributes such as upslope
area with a linear scaling thus enabling efficient calculation ($O(n)$) at each time step of the simulation even for large grids (Braun and Willett, 2013).

DEMs of real landscapes frequently contain data artifacts that generate topographic sinks. Topographic sinks can also occur as a result of diffusion on hillslopes by creating "colluvial wedges" damming the sections of the river network. By adopting algorithms of flow network derivation from TopoToolbox, TTLEM makes use of an efficient and accurate technique for
drainage enforcement based on auxiliary topography to derive non-divergent (D8) flow networks (Schwanghart et al., 2013; Soille et al., 2003). Based on the thus derived flow network, TTLEM uses downstream minima imposition (Soille et al., 2003) that ensures that downstream pixels in the network have lower or equal elevations than their upstream neighbors.

### 3.2. Tectonic displacement

We implement a 2D version of a flux limiting total volume method to reduce numerical diffusion when simulating tectonic
displacements on a regular grid. Equation (1) can be written as a scalar conservation law:

$$z_t + f(z)_x + f(z)_y = 0 \qquad (9)$$

where $f(z) = v_x z$ and $f(z) = v_y z$ are the flux functions of the conserved variable $z$. We refer to the supplementary material of Campforts and Govers (2015: Eq. SI 8 - 12) for a derivation of the differential form of Eq. (9) which can be converted to a numerical semi-conservative flux scheme:



$$z_{i,j}^{k+1} = z_{i,j}^k + \frac{\Delta t}{\Delta x}\left[ f_{i-\frac{1}{2},j} - f_{i+\frac{1}{2},j} \right] + \frac{\Delta t}{\Delta y}\left[ f_{i,j-\frac{1}{2}} - f_{i,j+\frac{1}{2}} \right] \qquad (10)$$

where $z_{i,j}^k$ is the elevation of the cell at row $i$ and column $j$ at time $k \times \Delta t$. $f$ represents the numerical approximation of the physical fluxes from Eq. (9). The in- and out coming fluxes are subsequently approximated with a flux limiting upwind method which is TVD. A TVD scheme prevents the total variation of the solution to increase in time and hence prevents spurious oscillations that are associated with higher order numerical methods. The use of a flux limiter allows the method to have a hybrid order of accuracy being second order accurate in most cases but shifting to first order accuracy near discontinuities. Hence the TVD-FVM method establishes a compromise between two desirable properties of a numerical method: it achieves a higher order of accuracy than first order schemes while ensuring numerical stability (Harten, 1983). TTLEM uses a staggered Cartesian grid for numerical discretization. The data grid points, or elevations from the DEM ($z$), are considered to represent the center of the computational cells, whereas the velocity fields ($v_x$ and $v_y$) are located at the cell faces.

The numerical TVD fluxes are calculated following Toro (2009):

$$f_{i+\frac{1}{2},j}^{TVD} = f_{i+\frac{1}{2},j}^{LO} + \varphi_{i+\frac{1}{2},j}\left[ f_{i-\frac{1}{2},j}^{HI} - f_{i+\frac{1}{2},j}^{LO} \right] \qquad (11)$$

where $f^{HI}$ and $f^{LO}$ represent the high and low order fluxes respectively:

$$f_{i+\frac{1}{2},j}^{LO} = \alpha_0 v_{i+\frac{1}{2},j} z_{i,j}^k + \alpha_1 v_{i+\frac{1}{2},j} z_{i+1,j}^k$$

$$f_{i+\frac{1}{2},j}^{HI} = \beta_0 v_{i+\frac{1}{2},j} z_{i,j}^k + \beta_1 v_{i+\frac{1}{2},j} z_{i+1,j}^k \qquad (12)$$

The low order fluxes are solved with a first order upwind Godunov scheme (1959):

$$\alpha_0 = \frac{1}{2}\left(1 + sign\left(\mathbf{v}\right)\right) \text{ and } \alpha_1 = \frac{1}{2}\left(1 - sign\left(\mathbf{v}\right)\right) \qquad (13)$$

The high order fluxes are solved with a Lax-Wendroff scheme (1960):

$$\beta_0 = \frac{1}{2}\left(1 + \mathbf{v}\frac{\Delta t}{\Delta x}\right) \text{ and } \beta_1 = \frac{1}{2}\left(1 - \mathbf{v}\frac{\Delta t}{\Delta x}\right) \qquad (14)$$

From Eq. (12), Eq. (13) and Eq. (14) it follows that:

$$f_{i+\frac{1}{2}}^{LO} = v_{i+\frac{1}{2},j} z_{i+1}^k$$

$$f_{i+\frac{1}{2}}^{HI} = \frac{1}{2} v_{i+\frac{1}{2},j}\left(z_i^k + z_{i+1}^k\right) - \frac{\left(v_{i+\frac{1}{2},j}\right)^2 \Delta t}{2\Delta x}\left(z_{i+1}^k - z_i^k\right) \qquad (15)$$

$\varphi_{i+\frac{1}{2},j}$ represents the flux limiter, which is solved with the van Leer scheme (1997):





$$\varphi_{i+\frac{1}{2},j} = \frac{r_{i+\frac{1}{2},j} + abs\left(r_{i+\frac{1}{2},j}\right)}{1 + abs\left(r_{i+\frac{1}{2},j}\right)} \qquad (16)$$

where $r$ is a smoothness index calculated as:

$$r_{i+\frac{1}{2}} = \frac{z_{i+2,j}^k - z_{i+1,j}^k}{z_{i+1,j}^k - z_{i,j}^k} \qquad (17)$$

The overall performance of the TVD-FVM is evaluated by comparing it with the first order accurate upwind Godunov scheme which is not flux limiting Eq. (13). In the remaining part of the text we refer to this scheme as the first order Godunov Method (GM).

### 3.3. River network updating

TTLEM features a 1D version of the flux limiting TVD-FVM to solve for river incision (Eq. (7)) which written as scalar conservation law is:

$$z_t + f(z)_x = 0 \qquad (18)$$

where $f(z)$ represents the flux function of the conserved variable $z$, representing the channel elevation. The method is similar than the one described in section 3.2 although fluxes are only calculated in one direction. We refer to the Supplementary Information provided by Campforts and Govers (2015) for a full derivation of this scheme. In addition, we implement a first

order explicit and implicit FDM for the solution of the stream power law detailed in Braun and Willett (2013). Implicit schemes provide stable solutions regardless of the time step considered, a property desired when simulating landscape evolution over long timescales and large spatial domains. An explicit scheme (both FDM and TVD-FDM), in turn, requires time steps that satisfy the Courant Friedrich Lewy condition (CFL):

$$\frac{KA^m \Delta t}{\Delta x} \leq 1 \qquad (19)$$

We introduce an inner time step ($\Delta t_{inner}$) for the simulation of river uplift and incision to achieve a sufficiently small time step

while maintaining an acceptable runtime (Fig. 1). TTLEM also allows for inner timesteps satisfying the CFL criterion if the implicit solution is used. While the implicit solution is unconditionally stable, an inner time allows us to investigate the impact of the length of the timestep on model outcomes (see section 5.1.2). Even when the Courant criterion is satisfied, model runs at low spatial resolutions can potentially allow very large timesteps. Large timesteps could imply a sudden input of vertical uplift in the solution resulting in the generation of artificial shockwaves. Therefore, TTLEM allows to user to set a maximum

length of the inner timestep ($\Delta t_{max}$) which we set by default to 3000 yr.

Regular grids introduce artefacts in the planform geometry of river networks because local drainage directions are restricted to eight directions (Braun and Sambridge, 1997). Moreover, as the process formulations are deterministic and flow direction algorithms follow a predefined order, LEMs tend to produce landscapes that are too uniform with respect to slope morphology and river planform patterns. To overcome this issue, we apply the method of Grimaldi et al. (2005) to explicitly integrate some

randomness in the calculation of the value of the drainage area exponent ($m$) by attributing a variance to $m$ :





$$\mathrm{var}(m) = \frac{\ln\left(1 + \frac{k_1}{k^2}\right)}{(\ln(A))^2} \qquad (20)$$

where $k_1$ and $k$ are proportionality coefficients. We update at each time step a new value of $m$ for each grid cell randomly drawing an error value from the distribution described by Eq. 13 and adding it to the mean value of $m$.

Another way to add variability in evolving landscapes is to allow the erodibility parameter $K$, to vary in space, thereby mimicking local, semi-random variations in rock strength. Here, variability on $K$ is simulated by introducing a normally distributed random deviation with a zero mean.

### 3.4. Hillslope processes

We implement linear hillslope diffusion using an efficient Crank-Nicolson scheme (Pelletier, 2008). This scheme is implicit and therefore allows large time steps. Implicit solutions are well suited since the diffusion equation is a parabolic PDE and much less sensitive to numerical diffusion in comparison to the stream power law, which is a hyperbolic PDE.

A numeric solution of the nonlinear hillslope equation is yet more demanding. The explicit FDM is limited by the maximum length of the time step at which numerical stability is maintained. Perron (2011) developed Q-imp, an implicit solver that allows to increase the length of the time step by several orders of magnitude. Whereas the per-operation computational cost of this algorithm is higher in comparison to the explicit solution, the overall performance of this method is better than hitherto alternative solutions (Perron, 2011). Q-imp efficiently calculates hillslope diffusion even for high-resolution simulations but is restricted to hillslopes below the threshold slope. Therefore, Q-imp must be combined with a hillslope adjustment algorithm.

We assume that hillslopes instantaneously adjust to oversteepening along fault scarps and due to river undercutting (Burbank et al., 1996). We refrain from simulating individual landslides although we acknowledge that single high magnitude low frequency events may be relevant at the time scales of our simulations (Korup, 2006). Instead, our approach implicitly accounts for the combined effects of a large number and variety of landslides that effectively adjust slopes to a threshold slope $S_c$. The threshold slope can be thought of "an average effective angle of internal friction which controls hillslope stability" (Burbank et al., 1996). We implement this hillslope adjustment using a modified version of the excess topography algorithm (Blöthe et al., 2015). In this algorithm, elevations $z$ at time step $t + 1$ are derived in a way that entails that the absolute local gradient at each grid cell is less or equal than $S_c$. This is achieved by decreasing elevations at locations $i$ to the minimum elevation of all other locations $j$ to which we add an offset calculated by the Euclidean distance $\|i,j\|$ and $S_c$:

$$z_i^{t+1} = \min\left\{z_i^t, \min\left[z_j^t + S_c\|i,j\|\right]\right\} \qquad (21)$$

The above equation entails that $z_i^{t+1}$ at one location depends on all other grid cells and that the algorithm has a time complexity of $O(N^2)$, which would render it unsuitable for frequent updating during LEM simulations. To avoid an overtly high computational load, we implement the algorithm using morphological erosion with a gray-scale structuring element (see MATLAB function ordfilt2), which is a minimum sliding window with additive offsets calculated from the window size and $S_c$. This significantly reduces run times as we calculate elevations at one location from the sliding window. Yet, this approach not necessarily removes all gradients greater than $S_c$. We solve this by calling the algorithm repeatedly until all slope values are equal or less than $S_c$.

We assume that albeit sediment might be temporarily redeposited in the system, it will be easily evacuated within a relatively short time span due to the unconsolidated nature of the deposits (McGuire and Pelletier, 2016). This assumption is reasonable




for rapidly uplifting and eroding mountain belts, but may not be applicable in other environments where mass wasting occurs (Vanmaercke et al., 2014).

### 3.5. Boundary conditions

TTLEM allows the use of Dirichlet or Neumann boundaries conditions. Alternatively, one can opt for a random disturbance at one or more boundaries of the modelled domain. The latter is especially of useful when simulating strong lateral

displacements which may otherwise generate artificially straight river profiles in the direction of the shortening.

## 4. Experiments

In order to demonstrate possible applications of TTLEM we carry out two series of numerical experiments. We first illustrate the impact of different hillslope process models on simulated landscape evolution, using a 30 m resolution DEM of the Big Tujunga region in California as an example. Second, we investigate the amount of bias and artificial symmetry introduced in

the landscape through the use of regular grids.

### 4.1. Hillslope processes

TTLEM allows to simulate hillslope processes assuming (non)-linear slope dependent diffusion with the consideration of a threshold hillslope. Figure 2 illustrates how different hillslope process algorithms affect the evolution of hillslopes in the Big Tujunga region, California (Fig. 2a). We assume no tectonic displacement and use standard parameter values for river incision

and hillslope diffusion (Table 1) and a threshold slope ($S_c$) of 1.2 (m/m) when applicable (Fig. 2b). We illustrate model results after 500 ky in Fig. 2c-d using the current topography as the starting condition. Linear diffusion (Eq. (4)) is not capable to keep up with river incision, which results in strongly oversteepened hillslopes near the river channels (Fig. 1c and 1g). While higher values for the diffusion coefficient $D$ will eliminate this problem (e.g. Braun and Sambridge, 1997) they are incompatible with experimental findings (Roering et al., 1999) and will restrict hillslopes to convex upward shapes. The use of non-linear

diffusion in combination with a threshold slope results in hillslopes similar to those simulated with linear diffusion in combination with a threshold slope. However, for a similar value of $D$, hilltops become more smoothed assuming non-linear diffusion as sediment fluxes due to diffusive processes now reach higher values when hillslopes approach the threshold slope.

### 4.2. Artificial symmetry

Regular gridded LEMs may introduce artificial symmetry in evolving landscapes (Braun and Sambridge, 1997). We perform

simulations with an entirely flat initial surface as well as with a random initial surface with uniformly distributed elevations between 0 and 50 m to investigate how random perturbations of the values of $m$ or $K$ affect drainage network evolution (Movie S1 and Movie S2). We consider four different scenarios for each initial surface (Fig. 3). Scenario 1 is the reference simulation, with a low spatial resolution of 1000 m, a large time step of $5 \times 10^4$ years and a $K$ value of $6 \times 10^{-6}$ m$^{-0.1}$yr$^{-1}$. In scenario 2, the mean erodibility $K$ is halved. In scenario 3 the time step is set to $1 \times 10^4$ years while in scenario 4, the spatial resolution is set

to 200 m.

At low spatial and temporal resolutions, the use of uniform parameter values results in clear artificial symmetry (Fig. 3). Introduction of random variability on $m$ mainly decreases similarity close to the river heads where the drainage areas are the smallest (scenario 1). This is a consequence of the formulation of Eq. (20): the introduced variability is relatively larger for small catchments. Variability in $K$ slightly decreases overall artificial symmetry at low spatial resolutions (scenario 1). The

use of a lower (mean) $K$ value, representing slower river incision also decreases overall artificial symmetry (scenario 2). Decreasing the time step (scenario 3) results in slightly different drainage networks in comparison to simulations with larger time steps but fails to reduce the symmetry in the result. At a high spatial resolution (scenario 4), artificial symmetry is still





present when constant parameter values are used. However, inserting variability on the *m* and *K* parameters is much more effective in reducing symmetry at this resolution.

Drainage networks simulated using an initial surface with elevations that randomly vary between 0 and 50 m are almost free of artificial symmetry and the final geometry of the drainage network is now less dependent of parameter variability. The latter underscores the importance of initial DEM conditions for the final results of a simulation (Perron and Fagherazzi, 2011). Nonetheless, even with a randomly varying initial surface, the perturbation on parameter values clearly affects the drainage network that is produced. Parameter value perturbation generally results in drainage networks which are less rectilinear than

those simulated without perturbation.

## 5. Impact of numerical methods

In a next step we investigate to what extent the numerical schemes implemented in TTLEM affect simulated landscape evolution. We distinguish between the effects on simulated river incision on the one hand and on simulated tectonic displacement on the other. We use a synthetically generated landscape for all simulations as a starting condition because we

are interested in the evaluation of the functionality of the model and not on the correct simulation of the evolution of a particular landscape or region. Hence, our simulations are uncalibrated and results were not compared with a 'true' landscape: however, the chosen parameter values are realistic.

### 5.1. River incision

#### 5.1.1.  1D river incision

The impact of numerical diffusion on propagating river profile knickpoints is most obvious in situations where an analytical solution is available. The first simulation illustrates such a situation, with an artificial river profile characterized by a major knickzone between 8 and 12 km from the river head (Fig. 4). We assume that the drainage area is increasing in proportion to the square of the distance and uplift equals zero. For this simple configuration, an analytical solution for the SPL can be found using the method of characteristics (Luke, 1972). Notwithstanding the relatively high spatial resolution of 100 m, both implicit

and explicit Finite Difference Methods (FDM) suffer from clear numerical diffusion when river incision is calculated over a time span of 1 Myr (Fig. 4). The TVD-FVM achieves a much higher accuracy, a finding that is systematic, occurring over a wide range of spatial resolutions and parameter values (Campforts and Govers, 2015).

#### 5.1.2.  River incision and catchment wide erosion rates

We hypothesize that apart from river profile evolution, the accurate simulation of river knickpoints will influence landscape

evolution as a whole. In order to investigate the sensitivity of catchment wide erosion rates to different numerical schemes of the river incision model, we first create a steady-state artificial landscape that we initialize with uniformly distributed random elevation values between 0 and 50 m on a 50 km × 100 km grid with a spatial resolution of 100 m (Movie S3). Landscape evolution is simulated using Dirichlet boundary conditions and by inserting spatially and temporally uniform vertical uplift of 1 km Myr$^{-1}$ over a period of 150 Myr. Outer model timesteps are set to $5 \times 10^4$ yr. Parameter values for river incision and

hillslope response are constant in space and time and are reported in Table 1. Figure 5 shows the resulting steady state landscape.

We impose four consecutive uplift pulses of equal magnitude to this artificial landscape (Fig. 5). Uplift pulses have a wavelength of 1.25 Myr and an amplitude of $1.5 \times 10^{-3}$ m yr$^{-1}$ (Fig. 6). TTLEM is run over 5 Myr with main model time steps of $5 \times 10^4$ yr, again with Dirichlet boundary conditions and planform fixed drainage network. We use two spatial resolutions

(100 m and 500 m) and three different numerical methods (implicit FDM without time step limitation, implicit FDM with time step limitation (CFL condition applied) and TVD-FVM) to simulate river incision. When applicable, the length of the inner





time step is set to $3 \times 10^3$ yr. Without inner time steps, river incision is calculated once for each main (outer) model time step ($5 \times 10^4$ yr).

We compare differences in simulated erosion rates by randomly selecting a number of catchments with drainage areas ranging
between 1 and 50 km$^2$ (221 and 202 catchments for runs at a spatial resolution of 100 m and 500 m respectively) (Fig. 8). We calculate the erosion rates for each time step by subtracting the elevation grid in the previous time step from the updated, current, elevation grid. The difference between the results obtained with different numerical schemes is quantified by calculating a Root Mean Square Error (RMSE):

$$RMSE = \sqrt{\frac{\sum_{i=1}^{n} \left(\varepsilon_{i,TVD} - \varepsilon_{i,FDM}\right)^2}{nb}} \qquad (22)$$

where $\varepsilon_{i,TVD}$ and $\varepsilon_{i,FDM}$ refer to the catchment wide erosion rates simulated with the TVD-FVM and FDM respectively to simulate river incision and $nb$ is the total number of discrete time steps of the simulated erosion record.

We rank the catchments from low to high RMSE for each comparison to investigate overall variations in catchment wide erosion rates. Figure 7 shows the results for the catchments at 10%, 50% (median) and 90% percentile. Note that the ranking is performed separately for the models runs at 100 m and 500 m as different sub catchments are randomly generated for both
simulation runs. The percentiles shown in Fig. 7 therefore represent different catchments.

For most catchments, we observe significant differences in erosion response between the three numerical methods at a spatial resolution of 100 m. The amplitude of the response to a tectonic uplift pulse increases when reducing numerical diffusion: the use of a first order implicit FDM without time step restriction results in a much smoother response in comparison to the TVD-FVM. The variations in response amplitude are significant: the majority of the catchments record amplitude reductions by
more 50% when modelled with the implicit FDM without time step restriction. Time step restriction (and thereby sacrificing the main advantage of the implicit FDM) significantly reduces numerical diffusion so that most catchments display an erosional response comparable to that simulated by the TVD-FVM. However, this finding is supported only by the simulation with 100 m spatial resolution. The advantage of a time step restricted implicit FDM over a non-restricted implicit FDM disappears almost completely for a coarser grid resolution of 500 m.

Catchment-wide erosion rates vary systematically with the use of different numerical methods. Figure 8 shows that erosion rates diverge between the different methods with increasing distance to the outlet of the main river while they are similar for larger catchments. A smaller effect of the numerical scheme on large catchment areas may be partly due to stronger averaging of local variations in catchments. In addition, catchments at a large distance from the outlet—and thus likely with smaller catchment areas—tend to experience the uplift signal only after several model time steps. If catchments are far from the fault
zone, knickpoints will then be significantly smoothed if an implicit FDM is used, which will affect the response of the catchment. This smoothening is not apparent if the catchment is close to the border of the modelling domain. Again, spatial resolution matters: a larger grid size not only results in larger differences on average but also in larger differences between small and large catchments (Fig. 8).

The differences in catchment response relate to the differences in simulated erosion rates within the catchments. Figure 9
illustrates the spatial difference in erosion rates calculated with the two numerical methods during the final step of the model run (after 5 Myr). This figure shows that spatial differences are significant and form a systematic banded pattern related to the upslope migration of the erosion waves of the individual uplift pulses.



### 5.2. Tectonic displacement

We test the performance of the 2D version of the flux limiting TVD-FVM to simulate tectonic displacement using a simplified model setup. We use a synthetic landscape as an initial condition and impose a constant lateral tectonic displacement while keeping erosion rates zero. Theoretically, this should result in a laterally displaced landscape that, apart from this, remains unchanged in comparison to the initial state. We compare the flux limiting TVD-FVM with a first order accurate upwind Godunov Method (GM). Figure 10 illustrates the results when applying a tectonic displacement in two directions ($v_x = v_y = 10$

mm yr$^{-1}$) over a time span of 1 Myr. The results show that the explicit GM strongly smooths the resulting DEM whereas the 2D TVD-FVM scheme produces a DEM that is very similar to the initial DEM, with minimal amounts of numerical diffusion.

In order to quantify and better understand the amount of numerical diffusion ($D_N$ [L$^2$ yr$^{-1}$]) introduced by the GM and the TVD-FVM method, we test a range of different model configurations and calculate the numerical diffusivity, $D_N$, corresponding to the observed smoothing. The latter is done by calculating the diffusivity required to transform the initial DEM ($DEM_{ini}$) to the

final DEMs produced at the end of the simulations ($DEM_{fint}$). The optimum amount of diffusion is determined by minimizing the misfit function $H$ with a sequential quadratic programming method (Nocedal and Wright, 1999). $H$ is given by:

$$H = \sqrt{\left(DEM_{ini} - DEM_{f\,int}\right)^2} \qquad (23)$$

Figure 11.a illustrates the relation between $D_N$ and the spatial resolution of different numerical approximations. The 2D TVD-FVM decreases numerical diffusion by a factor of 5-60 compared to the GM (Fig. 11b). The accuracy increases for both

schemes with increasing resolution and increasing CFL numbers. The increase in accuracy with higher spatial resolution is due to smaller spatial steps that result in better approximations of the spatial derivatives. Yet, the gain in accuracy with increasing spatial resolution is higher for the TVD-FVM than for the GM. Our analysis shows that the explicit FDM performs best with a CFL criterion close to one. This may seem counterintuitive as one might expect smaller time steps (CFL = 0.5) to lead to higher accuracies. However, the accuracy gain from an increase in temporal resolution is reduced by additional

numerical diffusion that is introduced by more iterations within a given time interval (Gulliver, 2007).

### 6. Discussion

There is a growing consensus that most eroding landscapes are in a transient state (Mudd, 2016; Vanacker et al., 2015). LEMs with high numerical accuracy are thus needed to capture transiency correctly, yet most commonly applied first order accurate

numerical methods introduce numerical diffusion and smear discontinuities that are inherent in transient landscapes. Knickpoints in river systems are of particular concern to geomorphologists as their analysis reveals insights into the tectonic and climatic controls on evolving landscapes. However, no analytical solution exists that allows to simulate river incision for changing drainage areas (Fox et al., 2014). Because drainage networks and drainage divides evolve in dynamic ways (Willett et al., 2014), the analysis of transient landscapes must thus rely on numerical methods, although analytical models can be

applied in specific cases (Perron and Royden, 2013). We present a higher order flux limiting scheme (referred to as TVD-FVM) that overcomes this problem of numerical diffusion.

Our simulations show that optimizing numerical schemes of LEMs is far from being only a numerical exercise. The impact of the numerical scheme to simulate detachment limited river incision on model outcomes is substantial and not limited to river profile development alone. Hillslopes adjust to local baselevel changes dictated by river incision. Hillslope denudation rates

must thus —at least partly— reflect the geometry of a knickpoint, whether it is a diffuse signal or a sharp discontinuity migrating upstream. Our simulations show that depending on the spatial and temporal resolution, catchment wide erosion rates are more responsive to uplift when derived from the TVD-FVM in comparison to FDMs. First order (explicit and implicit)



FDMs fail to properly reproduce transient incision waves (Campforts and Govers, 2015; Pelletier, 2008) with the effect that the smoothing propagates to inferred rates of hillslope denudation and that catchment wide erosion rates are smeared over geological time. Thus, the use of a shock preserving method such as TVD-FVM is strongly recommended for accurate simulations of transient landscapes.

It could be argued that TVD-FVMs are unnecessary as long as one applies an implicit method in combination with a sufficiently small time step. Although small time steps partly resolve the problem of smearing, their effect on numerical accuracy can hardly be generalized. Our simulations show that, for the selected parameter value combinations, results were only acceptable if a time step restriction is combined with a relatively high spatial resolution (100 m). In addition, it is well possible that, for other parameter value sets, numerical diffusion will be important, even if a fine grid is used. It would be infeasible for a model user to detect smearing problems in standard applications as comparable exact, analytically derived solutions, usually are nonexistent. Hence, we argue that the use of a shock capturing TVD-FVM numerical scheme is preferable since it avoids significant numerical diffusion under a wide range of parameter values and spatial resolutions. Moreover, by constraining the time step of a first order implicit method below the CFL criterion, the main advantage of an implicit scheme, i.e., the stability for any time step, disappears.

One might debate the significance and necessity of numerical schemes that avoid diffusion of retreating knickpoints. We think that it is critical to simulate knickpoint retreat using a method that avoids numerical diffusion. Even in bedrock-dominated landscapes knickzones are often smoothed, possibly due to flow acceleration above knickzone lips and subsequent localized higher erosion (Berlin and Anderson, 2007). The discrepancy between actual and simulated longitudinal profiles of hanging valleys has prompted (Valla et al., 2010) to prefer a transport-limited model (Willgoose et al., 1991a) over a detachment-limited model (Howard, 1994; Whipple and Tucker, 1999). The presence of significant sediment loads does not necessarily imply that transport limitations control river incision. Sediment flux dependent models, as first proposed by Sklar and Dietrich (1998) consider the hybrid role of sediment particles, acting as a tool to break and erode river beds in eroding regimes and as a covering armor in depositional regions (Gasparini and Brandon, 2011; Sklar et al., 1998). One-dimensional analytical simulations have shown that this process might generate over-steepened river reaches and explain the presence of permanent hanging fluvial valleys (Crosby et al., 2007). Numerical LEMs accounting for saltation-abrasion have so far not been able to reproduce such permanent hanging valleys: however this may be caused by the effects of numerical diffusion rather than by an inadequate process formulation (Crosby et al., 2007). Simulation of sharp knickpoints is also required in geomorphological and lithological settings where knickpoint retreat is caused by rock toppling, possibly triggered during extreme flood events, where knickpoint diffusion through abrasion and plucking of small blocks is minor (Baynes et al., 2015; Lamb et al., 2014; Mackey et al., 2014). Thus, various scenarios of knickpoint retreat exist, some which are characterized by significant natural diffusion, while others are not. In both cases simulation tools with a minimum of numerical diffusion are required to correctly quantify the importance natural diffusion.

First order numerical methods also inadequately simulate tectonic displacement on a regular grid. The amount of numerical diffusion that is introduced by these methods will, in many cases, far exceed natural diffusion rates, thus rendering accurate simulation of hillslope development impossible. A 2D variant of the TVD-FVM, instead, strongly reduces the amount of numerical diffusion ($D_N$) to values well below natural diffusivity values, an effect that is especially apparent at high spatial resolutions. We thus implemented a scheme that allows to accurately model a process that significantly impacts the evolution of topography and river networks (Willet, 1999), but whose simulation was hitherto mainly restricted to LEMs with flexible spatial discretization schemes.

Although most LEMs use first order accurate discretization schemes (Valters, 2016), the problem of numerical diffusion has been widely discussed in the broader geophysical community (Durran, 2010; Gerya, 2010). An alternative family of shock





capturing Eulerian methods being frequently applied are the MPDATA advection schemes (Jaruga et al., 2015). These schemes
        are based on a two-step approach in which the solution is first approximated with a first order upwind numerical scheme and
        then corrected by adding an antidiffusion term (Pelletier, 2008). However, contrary to the TVD-FVM, the standard MPDATA
        scheme (Smolarkiewicz, 1983) is not monotonicity preserving (or is not TVD). Instead, MPDATA introduces dispersive
        oscillations in the solution if combined with a source term (such as uplift) in the equation (Durran, 2010). Adding limiters to
the solution of the antidiffusive step (Smolarkiewicz and Grabowski, 1990) renders the MPDATA scheme oscillation free
        (Jaruga et al., 2015). However, by adding this additional correction, the method approaches the numerical nature of the TVD-
        FVM which does not require further adjustments in any case. Lagrangian schemes offer another alternative and are based on
        so called markers which evolve with the changing variable over time (Gerya, 2010). In the framework of a raster-based LEM,
        a fully Lagrangian tracing scheme is not desired and can be replaced by semi-Lagrangian methods that require interpolation
between the propagating markers and the grid cells (Spiegelman and Katz, 2006). These methods could potentially achieve
        high accuracy. However, simulation of horizontal topographic shortening would require large amounts of incremental markers
        to prevent numerical diffusion when interpolating the solution to the grid used in TTLEM. Both memory requirements and
        interpolation processing times therefore legitimize the use of the TVD-FVM which is sufficiently accurate and avoids
        interpolation.

Some of the weaknesses of the tested numerical solutions can be reduced by LEMs that rely on irregular grid geometries.
        TTLEM avoids these techniques but rather attempts to run on rectangular grids with a maximum of accuracy. We chose so for
        several reasons: First, input data such as topography, climate, lithology or tectonic displacement fields are typically available
        as raster datasets and thus require only minor modifications whereas irregular grids require substantial preprocessing. Second,
        TTLEM output can instantly be analyzed and visualized using the TopoToolbox library (Schwanghart and Kuhn, 2010;
Schwanghart and Scherler, 2014) or any other geographic information system. Thus, while irregular grid geometries and
        flexible grids may have some advantages over rectangular grids with respect to numerical accuracy, TTLEM's implementation
        of highly accurate algorithms strongly reduces the shortcomings of rectangular grids while facilitating straightforward
        processing of in- and output therefore enhancing the ease of modelling.

        TTLEM offers users the flexibility to address a number of issues. It allows users to define different initial conditions such as
a flat surface, a randomly disturbed surface or a DEM of a real landscape. TTLEM particularly benefits from the adoption of
        highly efficient drainage network algorithms that outscore GIS implementations in terms of computational efficiency while
        maintaining their ability to handle the artefacts (artificial topographic sinks) pertinent in real world DEMs (see Table 1 in
        Schwanghart and Scherler (2014)). TTLEM provides access to different models of hillslope denudation, and allows to model
        tectonic displacement at any desirable level of detail. Finally, TTLEM provides different numerical schemes to solve the
governing equations allowing users to trade-off between computational efficiency and accuracy. To our knowledge, such LEM
        versatility is hitherto inexistent and thus adds to the plethora of available LEMs (Valters, 2016). Its ability to be directly run
        on available DEMs renders TTLEM a simulation environment to explore trajectories of landscape evolution under different
        scenarios of geomorphological, climatological and tectonic controls.

### 7. Conclusion

TTLEM v1.0 is a raster based Landscape Evolution Model (LEM) contained within TopoToolbox. It allows using a flux
        limiting Total Variation Diminishing Finite Volume Method (TVD-FVM) to solve the stream power law and to simulate lateral
        displacements. The TVD-FVM solves river incision much more accurate which is reflected in catchment wide erosion rates.
        Depending on the spatial and temporal resolution used during model runs, first order implicit methods to simulate river incision
        lead to catchment wide erosion rates which are smeared out over the simulated time span and does not allow to properly capture
transient landscapes response. The fact that the impact of numerical schemes is not only altering simulated topography but





also simulated erosion records is of utmost importance in the light of the current debate where long term erosion histories are increasingly used to unravel uplift-erosion-climate dynamics. TTLEM features a range of hillslope response schemes to simulate hillslope processes and allows accurate simulation of lateral tectonic displacements, for example due to tectonic shortening. The combination of geomorphological laws to capture landscape response to changes in both internal (e.g. tectonic

configurations) and external (e.g. climate changes) forcings provides the community with a novel tool to accurately reconstruct, predict and explore landscape evolution scenarios over different spatial and temporal timescales.

**Code availability**

TTLEM 1.0 is embedded within TopoToolbox version 2.2. The source code and future updates can be downloaded from the GIT repository https://github.com/wschwanghart/topotoolbox. TTLEM is platform independent and requires MATLAB 2014b

or higher and the Image Processing Toolbox. Documentation and user manuals for the most current release version of TopoToolbox and TTLEM can be found at the GIT repository in the help folders of the software. The user manual of TTLEM includes three tutorials which can be accessed from the command window in MATLAB. To get started: download and extract the main TopoToolbox folder from the repository to a location of your choice. Add the folder to the Matlab search path by entering the following code in the command window *addpath(genpath('C:\path\...\TT_folder'))*. The software package comes

with three examples which can be initiated from the command window by entering *TTLEM_usersguide_1_intro* ; *TTLEM_usersguide_2_Synthetic_model_run* or *TTLEM_usersguide_3_Synthetic_Geological_Configuration*. These tutorials are also documented in the Help folder of ttlem. The source code for the solution of the one dimensional Stream Power Law (SPLM) can be downloaded from the GIT repository https://github.com/BCampforts/SPLM. SPLM contains the solution of the 1D river incision codes including four examples.

**Acknowledgements**

This work was motivated by the meeting "Landscape evolution modelling - bridging the gap between field evidence and numerical models" in Hannover, 21-23, October 2015, that was organized by the *FACSIMILE* network and funded by the Volkswagen Foundation. Additional support comes from the Belgian Science Policy Office in the framework of the Interuniversity Attraction Pole project (P7/24): SOGLO - The soil system under global change. Numerical simulations were

performed in the MATLAB environment (2015b) using numerical schemes as referred to in the text. We are grateful to the IDYST group of the University of Lausanne and in particular Frédéric Herman and Aleksandar Licul for inspiring discussions on numerical methods and Nadja Stalder for the figure design. We further thank Taylor Perron for sharing his source code.




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




**Figure 1.** Schematic representation of the TTLEM model flow. The numbered methods correspond with the paragraphs from
section 3 in the main text.


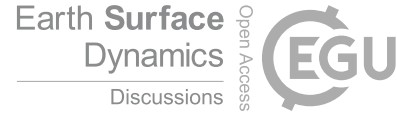

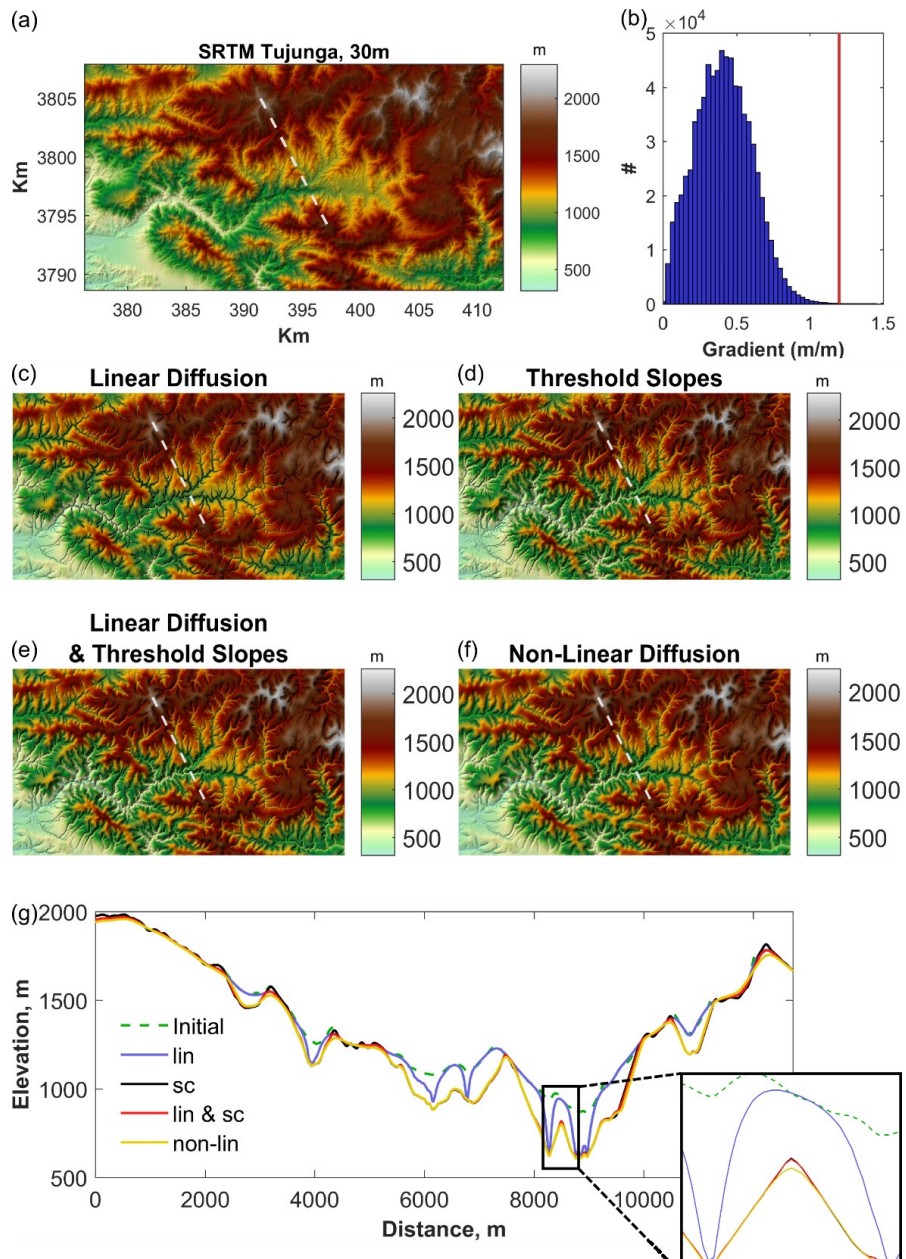

**Figure 2.** Hillslope response to river incision. (a) Standard SRTM DEM (30 m) included in TopoToolbox representing the Tujunga region. The dotted grey line indicates the location of the transect shown in subplot g. (b) Resulting topography after 500k years using four different descriptions for hillslope evolution. (c) Linear diffusion over all slope values (lin). (d) Threshold landscape where no slopes exceed the threshold slope (*Sc*). (e) Linear diffusion combined with immediate adjustment to a threshold slope (*Sc*). (f) Non-linear diffusion combined with immediate adjustment to a threshold slope (*Sc*). (g) Elevation profiles of the different model runs compared with the initial profile. Model parameter values are listed in Table 1.





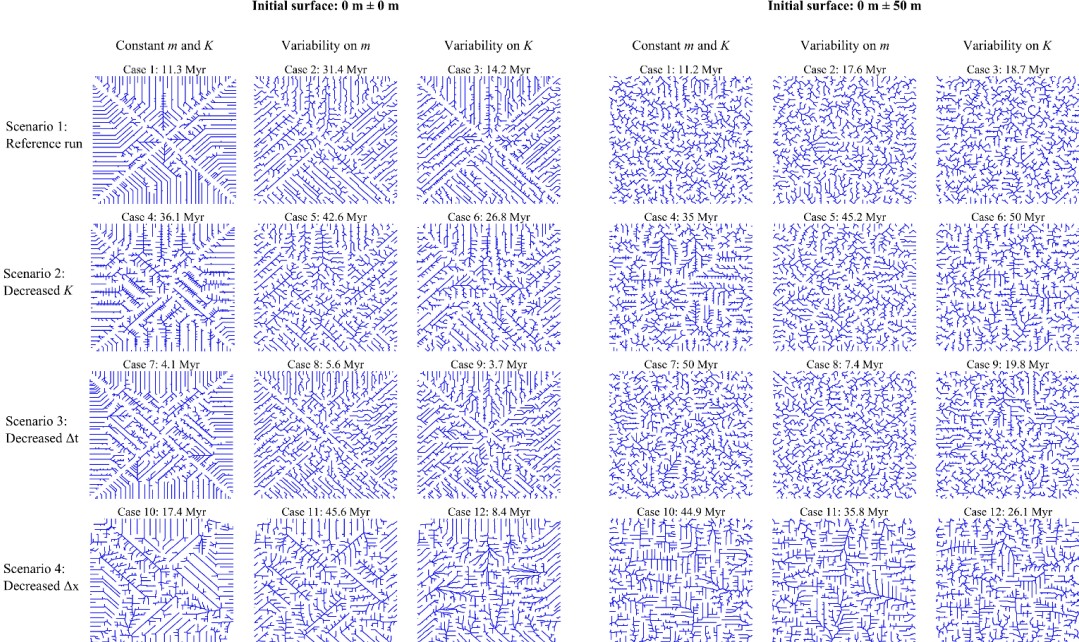

**Figure 3.** Steady state river networks obtained with different model configurations. The age at which a steady state is achieved is given in the title of the subplot. The first three columns in the left hand side of the figure represent model runs initiated from a flat, zero elevation surface. The first three columns in the right hand side of the figure represent model runs initiated from a surface with elevations randomly varying between 0 and 50 m. The configuration of the different model simulations is explained in the text and parameter values are listed in Table 1. Evolving river networks are presented in Movie S1 and Movie
S2.





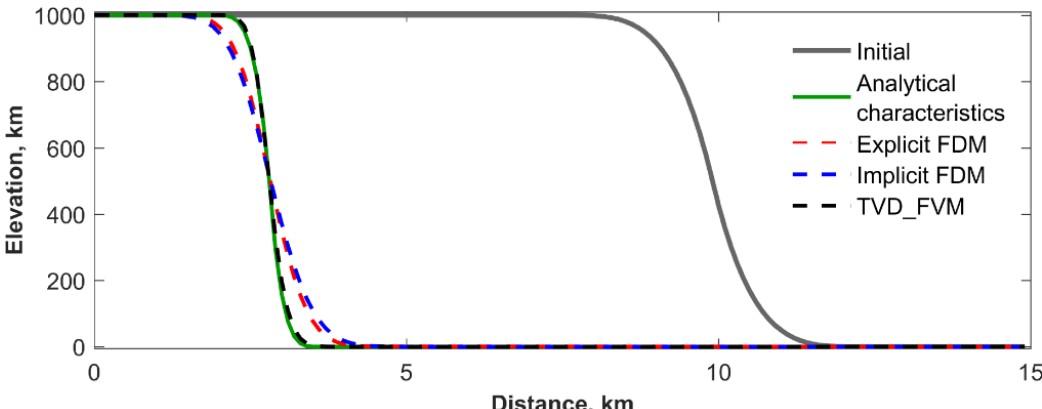

**Figure 4.** Solution of the linear 1D stream power law for a synthetic knickzone over a timespan of 1 Myr. The analytical

solution is obtained with the method of characteristics. The spatial resolution equals 100 m. Other model parameter values are

listed in Table 1.



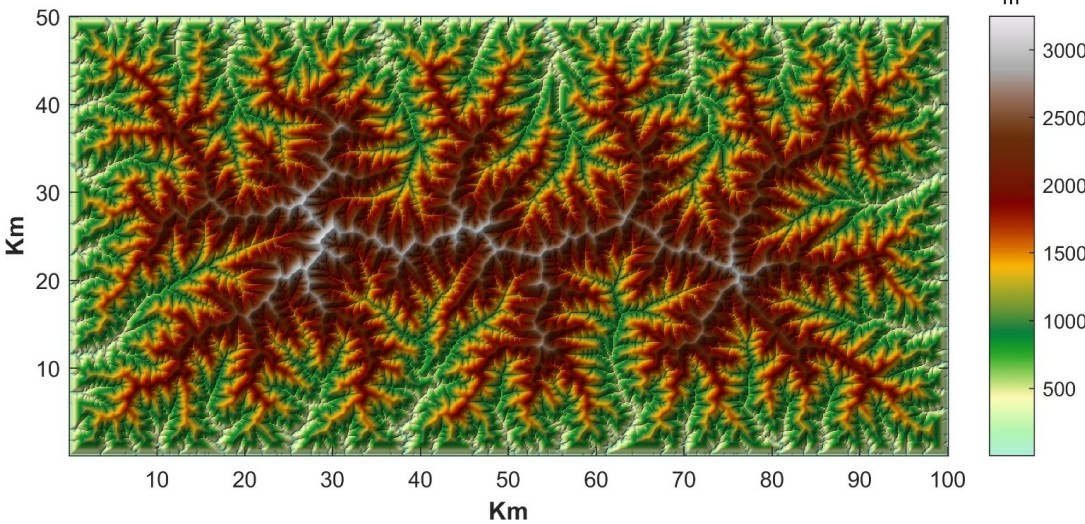

**Figure 5.** A synthetic steady state landscape produced as the testing environment to verify and compare the different numerical schemes implemented in TTLEM. Model runtime was 150 Myr, uplift rate was assumed to be spatially uniform over the area (block uplift) and fixed to $10^{-3}$ m yr$^{-1}$. Other model parameter values are listed in Table 1. Dynamic landscape evolution is presented in Movie S3.





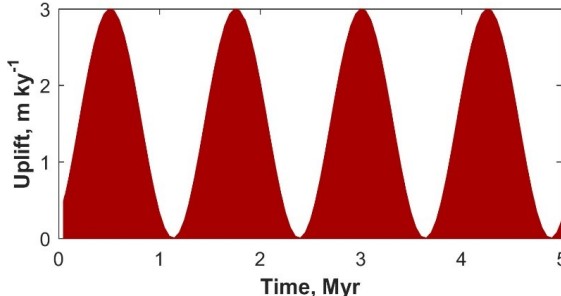

**Figure 6.** Uplift imposed to the steady state landscape show in Figure 5 to investigate the impact of different numerical schemes.





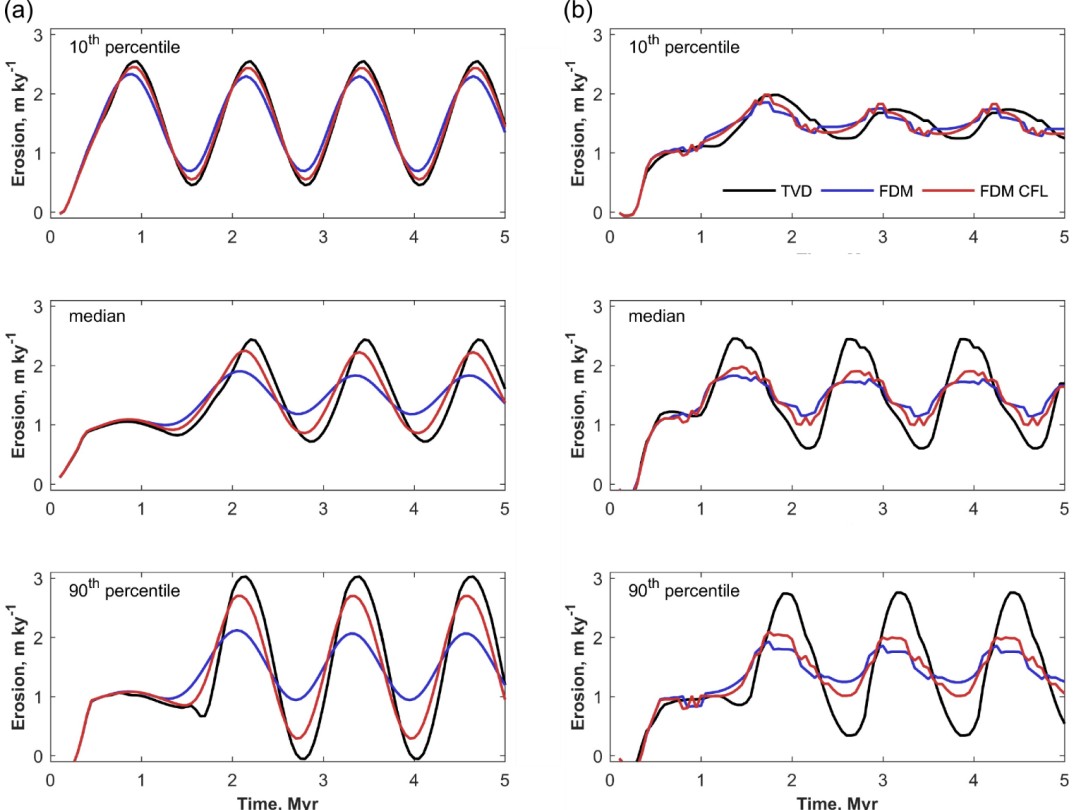

**Figure 7.** Temporal variation in simulated catchment wide erosion rates using different numerical methods to simulate river incision. The black lines represent simulations where a flux limiting TVD-FVM is used, the blue lines represent the implicit

FDM without constraints on the timesteps and the red lines represent the FDM with an inner timestep calculated with the CFL criterion. (a) Simulations performed at a spatial resolution of 100 m. (b) Simulations performed at a spatial resolution of 500 m. Here, a median filter with a window of 3 timesteps was applied on the simulated erosion rates to eliminate spikes which might occur at low resolutions.



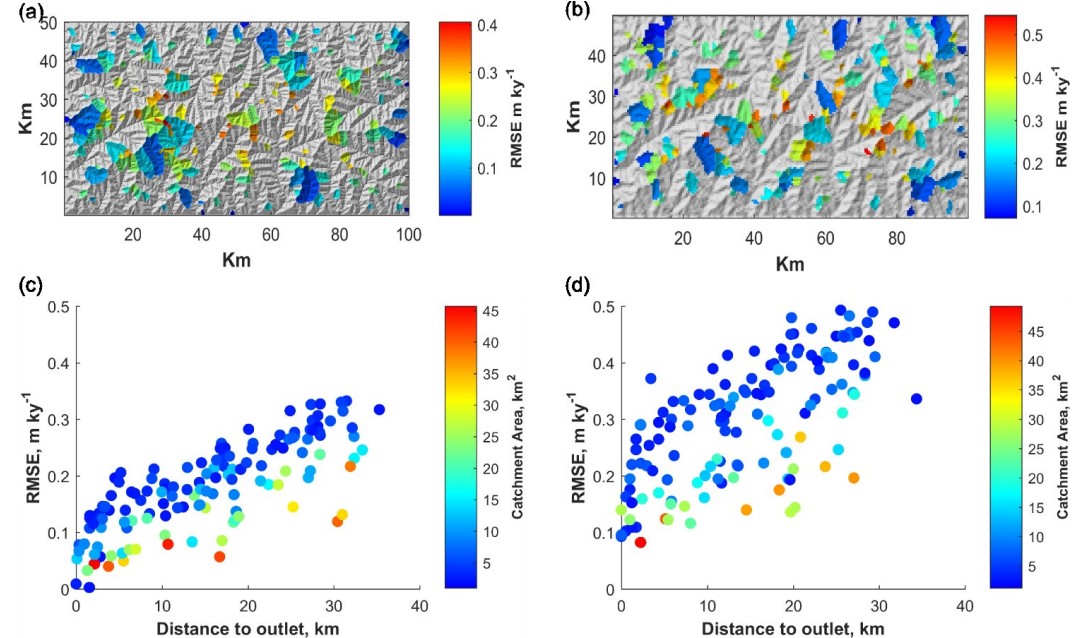


**Figure 8.** Spatial variation of differences between simulated erosion rates calculated with a flux limiting TVD-FVM for simulating river incision and an implicit FDM. Here, we compare methods both run with an inner timestep constrained with the CFL criterion (see text). RMSE is thus calculated between the black and red lines from Figure 7. Left column represents simulations run at a spatial resolution of 100 m, right column at 500 m. (a and b) Location of the randomly selected catchments

with an area > 1 km² and < 50 km². Colors refer to the RMSE between the two simulations. (c and d) Differences between the schemes increase with increasing distance from the river outlets and are inversely correlated with the catchment area.





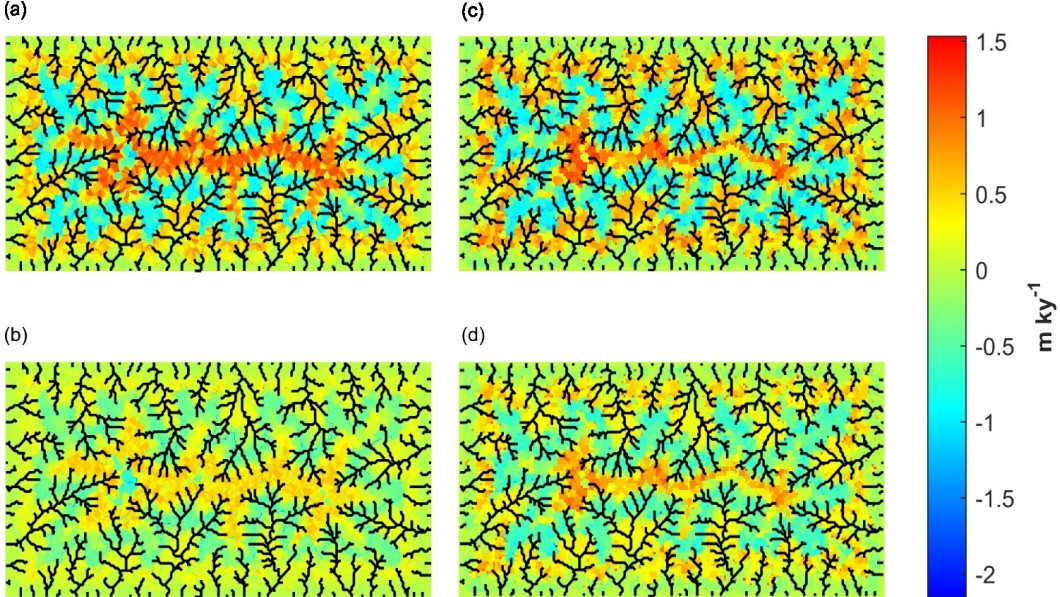

**Figure 9.** Spatial pattern of erosion rates during one model timestep when simulating landscape evolution with the flux limiting
TVD-FVM versus the first order implicit FDM. (a) simulation at a resolution of 100 m where the timestep of the implicit
method is not constrained (b) simulation at a resolution of 100 m where the timestep of the implicit method is constrained with
the CFL criterion (c) simulation at a resolution of 500 m where the timestep of the implicit method is not constrained (d)
simulation at a resolution of 500 m where the timestep of the implicit method is constrained with the CFL criterion.






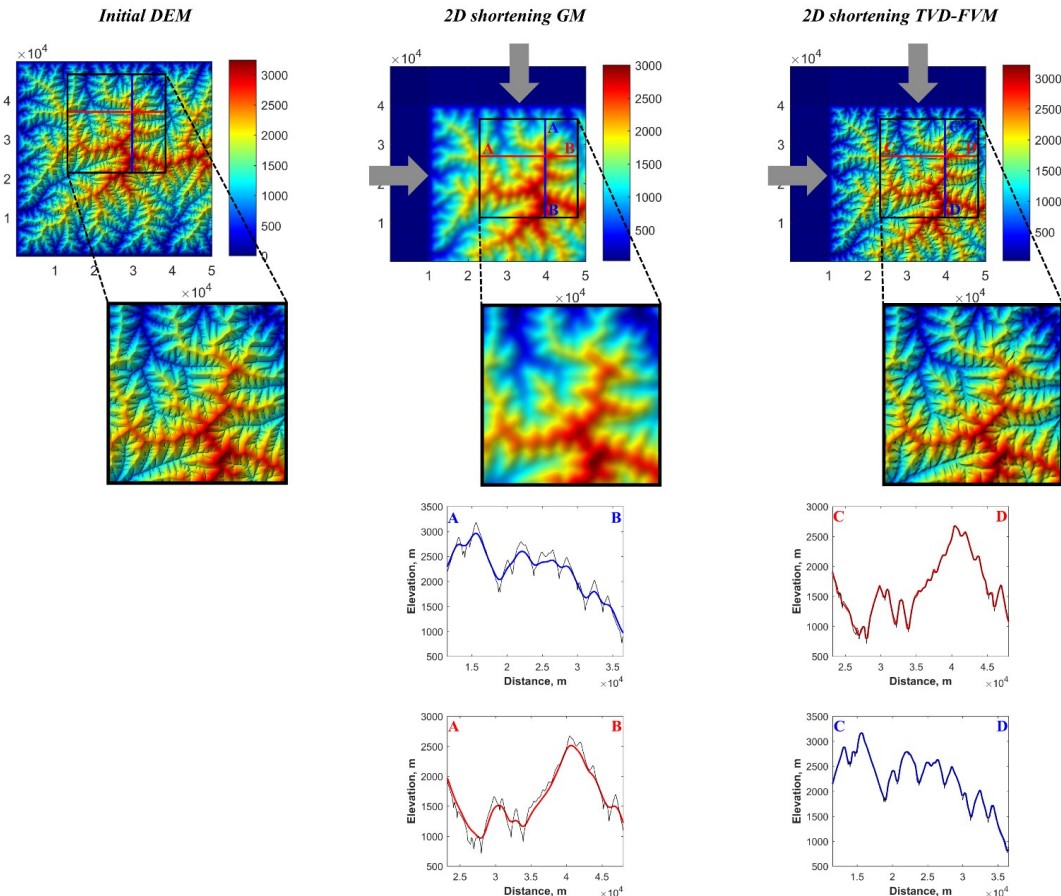

**Figure 10.** Impact of numerical schemes when simulating horizontal shortening on a fixed grid. Left: extract from synthetically produced DEM from Fig. 5. Middle: horizontal shortening in two directions simulated with a 2D explicit first order Godunov Method (GM). Right: horizontal shortening in two directions simulated with a 2D explicit flux limiting TVD-FVM.





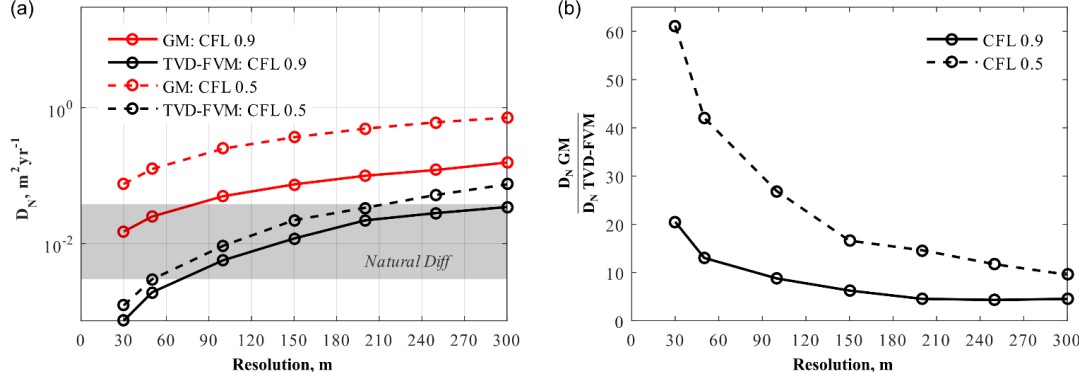


**Figure 11.** (a) Amount of numerical diffusion ($D_N$) introduced in the system when simulating lateral tectonic displacement in two directions as a function of raster resolution. The grey zone indicates the range of naturally observed diffusion rates. (b) The ratio between the amount of numerical diffusion for the first order Godunov Method (GM) versus the flux limiting TVD-FVM.



**Table 1.** Model parameters used for the TTLEM simulations.

| Parameter | Units | Figure 2 | Figure 3 | Figure 4 | Figure 5 | Figure 7-9 |
|---|---|---|---|---|---|---|
| | | | *Initalization* | | | |
| InitialSurface | | Tujunga SRTM | flat - random | flat, 1D | random | synthetically produced DEM shown in Fig. 3 |
| UpliftPattern | | - | uniform | variable in time | uniform | uniform |
| UpliftRate | m yr$^{-1}$ | 0 | $1 \times 10^{-3}$ | $0 - 1 \times 10^{-3}$ | $1 \times 10^{-3}$ | $0 - 3 \times 10^{-3}$ |
| SpatialStep | m | 30 | 200 - 1000 | 100 | 100 | 100 - 500 |
| | | | *Computational parameters* | | | |
| TimeSpan | yr | $5 \times 10^5$ | $50 \times 10^6$ | $1 \times 10^6$ | $150 \times 10^6$ | $5 \times 10^6$ |
| TimeStep | yr | 1250 | $1 \times 10^4$ - $5 \times 10^4$ | ca. $5 \times 10^3$ | $1 \times 10^5$ | $5 \times 10^4$ |
| AreaThresh | m$^2$ | $2 \times 10^5$ | $2 \times 10^6$ | - | $2 \times 10^5$ | $2 \times 10^5$ |
| DrainDir | | variable | variable | - | variable | - |
| DiffToRiv | | FALSE | FALSE | - | FALSE | - |
| steadyState | | FALSE | TRUE | - | TRUE | - |
| SS_Value | m | - | 5 | - | 0.5 | - |
| parallel | | FALSE | FALSE | - | FALSE | - |
| massWasting_river | | FALSE | FALSE | - | FALSE | - |
| | | | *Boundary conditions* | | | |
| BC_Type | | Neumann | Dirichlet_Matrix_Ini | - | Dirichlet | - |
| BC_dir_DistSites | | - | - | - | - | - |
| BC_dir_Dist_Value | | 1 | 1 | - | 1 | - |
| BC_dir_value | | 0 | 0 | - | 0 | - |
| BC_nbGhost | | 1 | 1 | - | 1 | - |
| FlowBC | | - | - | - | - | - |
| | | | *River incision* | | | |
| Kw | L$^{1-2m}$ t$^{-1}$ | $4 \times 10^{-6}$ | $6 \times 10^{-6}$ - $3 \times 10^{-6}$ | $5 \times 10^{-6}$ | $5 \times 10^{-6}$ | $5 \times 10^{-6}$ |
| m | | 0.45 | 0.45 | 0.42 | 0.45 | 0.45 |
| n | | 1 | 1 | 1 | 1 | 1 |
| m_var | | 0 | 0 - 0.2 | - | 0 | - |
| K_weight | | - | 0 normally distributed | - | - | - |
| NormPrecip | | - | - | - | - | - |
| | | | *Hillslope response* | | | |
| D | m$^2$ yr$^{-1}$ | 0.015 | 0.036 | - | 0.036 | 0.036 |
| ρr ρs-1 | - | 1.3 | 1.3 | 1.3 | 1.3 | 1.3 |
| DiffTol | | $1 \times 10^{-4}$ | $1 \times 10^{-4}$ | - | $5 \times 10^{-4}$ | $1 \times 10^{-4}$ |
| Sc | m m$^{-1}$ | 1.2 | 0.7 | - | 0.7 | 1 |
| Sc_unit | | tangent | tangent | - | tangent | - |
| | | | *Tectonic shortening* | | | |
| shortening | | FALSE | FALSE | - | FALSE | - |
| short_x | m yr$^{-1}$ | - | - | - | - | - |
| short_y | m yr$^{-1}$ | - | - | - | - | - |
| | | | *Numerics* | | | |
| riverInc | | implicit_FDM | implicit_FDM | implicit_FDM TVD_FVM | implicit_FDM | implicit_FDM TVD_FVM |
| cfls | | 0.7 | 0.7 | 0.7 | 0.7 | 0.7 |
| diffScheme | | imp_lin only_sc imp_lin_sc imp_nonlin_sc | imp_lin_sc | - | imp_lin_sc | - |
| shortening_meth | | Upwind_TVD | Upwind_TVD | - | Upwind_TVD | - |
| | | | *Model output* | | | |
| ploteach | | 1 | 1 | - | 1 | 0 |
| saveeach | | 1 | 1 | - | 1 | 0 |
| fileprefix | | res_ | ArtSym_RandIni_1_SimNb_1_ | - | standard_run_100m | - |
| resultsdir | | C:\DATA_... | C:\DATA_... | - | C:\DATA_... | - |