# Peer review of "Accurate simulation of transient landscape evolution by eliminating numerical diffusion: the TTLEM 1.0 model"

_Earth Surface Dynamics, 2016_

## Referee Comment (RC1) · Anonymous Referee #1 · 6 Aug 2016

This paper is an extension of a recent contribution by Campforts and Govers (2015) that demonstrated the efficacy of using a higher-order flux limiting total volume method (TVD-FVM) for modeling the advective (i.e., stream power law) component of a coupled hillslope-fluvial landscape evolution model. The authors have extended the TVD-FVM method to 2D and they are making the new LEM available to the community as TTLEM.

The main point of this paper is absolutely correct: that upwind differencing with no correction introduces significant numerical diffusion into LEMs. The conclusion that upwind differencing without correction is unacceptably diffusive can be found in every numerical modeling textbook of the last few decades. I don't point this out to minimize the important contribution that the authors have made. Rather, I agree with them

that upwind differencing is overly utlized in the LEM community, often without scrutiny. About this there should be no debate. It should be noted that the numerical diffusion introduced by upwind differencing can be computed and may, in some cases, be mitigated by reducing the diffusivity coefficient D by the same amount introduced by upwind differencing, but this work-around is not commonly performed and is only possible if the prescribed value of D is sufficiently large. I applaud the authors for highlighting the problem of numerical diffusion (first in Campforts and Govers (2015), and again here) and for proposing a robust solution to the problem.

That said, I think the tests employed by the authors do not always allow for a clear assessment of the advantages of TVD-FVM. The authors make comparisons between a first-order upwind method and a higher-order TVD method for computing fluxes. However, unless I have misundertood something, the time steps used are variable within the models, making it difficult to clearly compare the errors associated with temporal discretization and clearly separate them from errors associated with spatial discretization.

Before I discuss this issue further, I think it is important to note that LEMs, like solutions to any other PDE or set of PDEs, should converge as the pixel size goes to zero, or at least be relatively insensitive to the grid resolution over the range of resolutions to which the model is applied. Without this, there is no unique solution for a given set of parameter values, making it impossible to know, in the absence of an analytic solution, if one has achieved the correct solution or to objectively compare results obtained with different schemes (the focus of this paper). Moreover, if a LEM is grid-resolution dependent then the same numerical model operating at different resolutions has to be separately calibrated to data, rendering parameter values such as D and K that should be solely functions of natural processes and material properties also functions of grid resolution. Pelletier, Geomorphology, (2010) has provided some guidance on how to make coupled hillslope-fluvial LEMs grid-resolution independent. His approach involves reframing the stream power as unit stream power (following all sediment transport formulae ever proposed, which is not a trivial rescaling since the contributing area generally scales with the pixel size on planar hillslopes but is relatively independent of the pixel size in convergent portions of the landscape) and modifying the strength of the diffusion term to account for the fact that changes in cross-sectional slope at valley bottoms occur over a distance equal to the valley bottom width (a property of nature), not the pixel size (not a property of nature). The random component of the model used by Campforts et al. poses a special challenge to achieving grid-resolution independence. However, one can maintain grid-resolution independence in a model with spatial random variability by generating random field(s) sampled at a resolution that represents the largest resolution the model will be applied to, then bilinearly interpolating these fields for use in versions of the model run at higher resolution. I am not suggesting that the authors adopt all (or any) of these suggestions, but I do suggest that this issue needs to be addressed in some way. The error calculation (equation (22)) simply assumes that the solution with TVD-FVM is exactly correct and any difference from this solution is an error. Without establishing grid-resolution independence it is really impossible to tell whether outputs such as Figures 8A and 8B are even unique solutions for a given set of parameter values, much less which one is more accurate.

The different methods are only evaluated for a small number of cases (two grid resolutions and cases with and without a maximum time step). Error in a first order method will decrease linearly as you decrease dx and it will decrease with $(dx)^2$ for a second order method. In moving from a grid with dx=500m to dx=100m, there is a large difference in the computed values of E depending on whether or not a first-order or higher-order method is used. This is expected, but this doesn't indicate a fundamental problem with any of the numerical methods. The error associated with each of the methods is dependent on the grid resolution. So, it is a given that there will be some range of grid resolutions where the differences between a 1st order and 2nd order method appear unacceptable (i.e. numerical diffusion is excessive relative to the prescribed diffusivity). However, what really matters in judging method accuracy is the computational time required to reach a given level of accuracy relative to an exact/converged solution. What would be most helpful is to demonstrate that TVD-FVM saves considerable computational time by providing an acceptable solution at a much higher grid resolution and/or is robust for a much wider range of grid resolutions than first order methods. I suggest the following: First, for one method, perform the simulation for a range of grid resolutions (400m, 200m, 100m, 50m, 25m,12.5m) until the solution converges, i.e. becomes essentially grid-resolution independent. Use a time step that is small enough so that the solution does not depend on the time step (this probably means using a time step that yields a very low Courant number for the coarser grids, but the magnitude of the time step is likely to be similar to the magnitude of the time step needed to keep the model stable on finer grids). Then, it is easy to argue that most of the error introduced into the solution is associated with the spatial component of the problem. Second, repeat step 1 for each of the numerical methods. Assuming all simulations are run on the same machine, keep track of the time required to perform the simulations. This would allow for a more robust comparison of the different methods and would give readers a better idea of the true differences between the methods. For instance, the TVD method should converge to an grid-resolution-independent solution more quickly than the lower order methods. But how much faster? How does this depend on uplift rate or other commonly varied parameters? What are the practical implications in terms of computing time? This would give readers more guidance on the necessity of using one method over the other.

In the discussion the authors imply that their method is really the only acceptable method for the stream-power component of LEMs. Techniques that are widely used to prevent artificial numerical diffusion in many fields of science, including MPDATA and semi-Lagrangian techniques, are implied to be inferior or less robust with no evidence. For example, semi-Lagrangian methods are deemed to be potentially of higher accuracy, but then simply dismissed as inferior to TVD-FVM because "simulation of horizontal topographic shortening would require large amounts of incremental markers to prevent numerical diffusion when interpolating the solution." This sentence confuses two different methods (semi-Lagrangian and particle-in-cell methods are not the same)

and is not based on any evidence. I don't see any point in discouraging the community from trying alternative methods until they are clearly tested and shown to be inferior for a wide range of potential applications

Minor issues:

1) The variable x is used for two different things (in eqn. (1) it represents one of the cardinal horizontal directions but in eqn. (2) is represents the along-channel distance).

2) There is some repetition and inconsistency in the equations. For example, there are 6 different equations for one variable (dz/dt). It would be better to use a notation that differentiates among different aspects of dz/dt (tectonic advection versus diffusive erosion/aggradation versus stream-power-driven erosion) and make it clear that dz/dt is the sum of these different components. As written, equations (1) and (6) and (9) are repetitive and incompatible, because they are almost the same equation, yet the left hand side of all of the equations is the same while the right hand side includes uplift in one of the equations but not in the other two.

3) It would be helpful for the authors to address whether the method could be applied to the nonlinear stream power law (n not equal to 1), spatially variable K (e.g., strong over weak layers in sedimentary or metamorphic rocks), transport-limited fluvial processes, landscapes with a finite soil layer over bedrock or intact regolith, and other common LEM variants.

4) The paper is comprehensively referenced, which I appreciate, but some of the references do not support the points being made. To take one example, McGuire and Pelletier (2016) is used to defend the use of a detachment-limited model on the basis that unconsolidated sediment can be easily evacuated from the fluvial network. This is simply untrue. Unconsolidated sediments obviously do get stored in fluvial systems. Whether a detachment-limited model is a reasonable approximation depends on the application (including details such as mean grain size), and I don't think a paper that deals with small channels forming on alluvial terraces is an appropriate basis for

defending the use of a detachment limited model in an LEM designed to model the large-scale evolution of mountain belts.

5) The structure of the paper is good but the sections/subsections could be slightly improved. For example, the issue of artificial symmetry that can arise with rectangular grids is first introduced on line 206 with no prior mention or subsection break. I think this issue should be addressed in its own subsection of section 3 (as it is in section 4.2).

6) The stream power model is introduced using its nonlinear form (the exponent n is general) but the remainder of the paper, including the CFL ondition (eqn. (19)), applies only to the linear case.

7) The use of D8 routing seems unsubstantiated. Dinfinity is the choice of nearly every modern LEM, because it more faithfully represents flow on hillslopes.

8) Please use lat/lon or UTM coordinates in Fig. 2. If these are UTM coordinates, please specify.

9) The method of the paper is referred to as TVD-TVM throughout the abstract but TVD-FVM in the paper. If this is not a typo, please explain the difference between these abbreviations.

10) $w\_A$ and $w\_k$ are introduced in the equation but then (unless I missed it) never discussed again (not even in the table of parameter values).

---

## Referee Comment (RC2) · Anonymous Referee #2 · 17 Aug 2016

Campforts et al. addresses an important problem for fluvial landscape evolution models: numerical diffusion of the solution to the stream power advection equation. The authors first of all present a solution to the problem based on a higher-order flux-limiting method (TVD-TVM), and secondly, they outline a new modeling platform (TTLEM), which makes use of TVD-TVM and is available to everyone as part of the TopoToolbox 2.0.

Overall, my opinion is that numerical accuracy of fluvial landscape evolution models has received too little attention in the past, and it is therefore good to see the authors address it here. The method proposed to reduce numerical diffusion is convincing, and the damping of numerical diffusion in stream-power advection as well as in tracking horizontal tectonic displacements is significant. I hope that this contribution gets published in Esurf, although I do have some concerns and suggestions, which I list below:

General comments:

First of all, I think the duel purpose of the manuscript: 1) discussing numerical diffusion and presenting TVD-TVM, and 2) presenting TTLEM as a more general landscape evolution model leads to a rather diffuse and ackward structure of the text. The main strength of this text is in my opinion the focus on numerical diffusion and the presentation of TVD-TVM, but the TTLEM presentation calls for many details that are not needed to address this issue (see for example Fig. 1). For example, because the introduction focuses mostly on the influence of numerical diffusion, it is hard to understand the motivation for the first couple of experiments focusing on drainage networks and the influence of different hillslope models. I would strongly recommend simplifying the flow of the manuscript focusing more exclusively on the issue of numerical diffusion. Likewise the authors should consider skipping the first two experiments and in stead perform more like the one shown in Fig. 7. I think that it would increase the impact of the contribution, and the presentation of TTLEM could perhaps be saved for another manuscript in a more software-oriented journal.

Secondly, I suggest the authors give a short introduction to basic knowledge about numerical diffusion in advection problems. This could be inspired by simple textbook material and use linear advection as a starting point. By this the authors could avoid some awkward reflections, like in line 378: it is not at all counterintuitive that time steps smaller than the CFL criterion leads to more numerical diffusion. Most numerical analysis textbooks I know of give very simple explanations for why numerical diffusion is minimized exactly at the CFL criterion. Overall, I think the authors can make better use of basic textbook wisdom to prepare the reader for the main points of the manuscript.

Finally, while I fully appreciate the comparison experiments between the different numerical methods, I suspect that it is not completely fair. The main advantage of the implicit method (as FastScape by Braun and Willett) is that it becomes more compute efficient at high spatial resolution than the explicit methods, simply because it is not similarly constrained by the CFL condition. Thus, if explicit and implicit methods were compared in experiments with similar compute time (which I think they should be), would the implicit method not allow for finer spatial resolution than the explicit method? If so, would the finer spatial resolution in combination with the larger time steps not reduce the numerical diffusion of the implicit method? I am not questioning the advantages of TVD-TVM here. I just feel that the authors are not appreciating the real strength of the implicit method, which is how the compute time scales with spatial resolution.

More specific comments:

Line 30: "availability of potential energy"

Line 85: delete "most"

Eqn 1: Why are vx and vy bold?

Eqn 2: Are wk and wa used for anything here? If not flush them out.

Line 102: what is "eroding settings"?

Eqn 3: The diverge operator should include a dot between nabla and qs

Line 103: hillslope erosivity and erodibility. What is the difference?

Eqn 7: Again, is the variability on m really needed to demonstrate the points of numerical diffusion? If not skip it to clean the text. More complicated means less convincing.

Eqn 8: I do not understand the effect of densities here. Is U not simply uplift of the surface? If so, I guess the densities should be on the second term, right?

Line 153: "...transforms returns..."

Eqns 11-17: The use of subscripts seems inconsistent.

Line 192: ".. is similar than the one. . ."

Eqn: 19: I guess A varies by several orders of magnitude in the grid. Please discuss the CFL criterion in the light of this. Is max(A) used here?

Line 199: Description of the inner time step is confusing, and I do not understand why it is needed. Again I suspect that it is the general presentation of TTLEM that stands in the way for a clear and concise presentation of the numerical experiments.

Lines 206-205: This kind of randomness should be avoided here. The authors are documenting the level of numerical diffusion in different numerical techniques, and in this process it is very important that we know what advection equation is solved. m seems to be varied in order to make the drainage networks look more realistic. But that is not important here. And by the way: varying m randomly does not remove the grid dependency (which is inherent to stream-power advection and D8 drainage), it just obscures the close links between the grid, the (random) variability of m, and the drainage network. Please keep m fixed and the equations as simple as possible!

Section 3.4 is not well written. In spite of carefully reading the text I am still confused about how hillslope processes are implemented. But more importantly: Can the experiments documenting numerical diffusion not be run without hillslope processes? This would require that Ac=0 in Eqn 8, but why not? It seems a bit silly to deliberately add physical diffusion to an experiment were one wants to measure numerical diffusion? The authors should consider if the experiments can be made simpler (see first general comment above). Skipping hillslope processes and deleting this section could be a quick fix.

Section 4: I recommend skipping the first two experiments on hillslope processes and drainage networks (or save them for another paper). This would free up space to dig deeper into advection and numerical diffusion.

Line 276: I am not impressed by this strategy. I agree that the artificial symmetry is a problem, but at least we know where it comes from. Fixing this by introducing variability in the exponent m obscures the link between model input and model output, which is otherwise critical for use of computational experiments. Variability on K is better, because the linear scaling does not alter the form of the equation.

Line 344: So, what happens if the grid resolution is lowered to 10 m?

Line 391: overcomes -> reduces

Line 403: A small time step is not the essential factor here. The implicit method first of all offers a fine spatial resolution in combination with a large time step. The advantage of this combination should be explored more.

Line 464-474: All of this seems rather irrelevant to the main points of this study. see first general comment.

Line 481: "... the current debate..." calls for references.

Fig. 1: I almost get dizzy by looking at this. What is the point of showing this level of complexity in the first figure?

Fig. 2: While this is interesting I do not understand the motivation. The introduction spins me up to read about numerical diffusion, not this.

Fig. 3: Same comments as for Fig. 2.

Fig. 4: This is a nice, simple figure and to me the extension of this existing result to 2D simulations is the essential contribution of this study. This figure could be a great opening figure.

Fig. 7: If the authors choose to follow my advise and skip the first experiments, then more like this could be performed. It would be useful to see experiments with different setting of m and n (linear vs. nonlinear). Also to have experiments at finer spatial resolution where the advantages of the implicit method should start to kick in.

**ESurfD**

Interactive
comment

Fig. 9: It is good to see the difference between methods here, but it would also be great to see pictures of the two separate erosion rates. I wonder if knickpoints can be recognized in both?

Fig. 10: great figure

---

## Author Response (AR1)

**Response to referees for: Accurate simulation of transient landscape evolution by eliminating numerical diffusion: the TTLEM 1.0 model**

We thank referee 1 for her/his comments, which helped improving the quality of the manuscript. Our responses the comments are in blue.

This paper is an extension of a recent contribution by Campforts and Govers (2015) that demonstrated the efficacy of using a higher-order flux limiting total volume method (TVD-FVM) for modeling the advective (i.e., stream power law) component of a coupled hillslope-fluvial landscape evolution model. The authors have extended the TVD-FVM method to 2D and they are making the new LEM available to the community as TTLEM. The main point of this paper is absolutely correct: that upwind differencing

with no correction introduces significant numerical diffusion into LEMs. The conclusion that upwind differencing without correction is unacceptably diffusive can be found in every numerical modeling textbook of the last few decades. I don't point this out to minimize the important contribution that the authors have made. Rather, I agree with them that upwind differencing is overly utilized in the LEM community, often without scrutiny. About this there should be no debate.

It should be noted that the numerical diffusion introduced by upwind differencing can be computed and may, in some cases, be mitigated by reducing the diffusivity coefficient D by the same amount introduced by upwind differencing, but this work-around is not commonly performed and is only possible if the

20 prescribed value of D is sufficiently large. I applaud the authors for highlighting the problem of numerical diffusion (first in Campforts and Govers (2015), and again here) and for proposing a robust solution to the problem.

**We are grateful for the appreciation of the reviewer regarding our work.**

1. That said, I think the tests employed by the authors do not always allow for a clear assessment of the advantages of TVD-FVM. The authors make comparisons between a first-order upwind method and a higher-order TVD method for computing fluxes. However, unless I have misunderstood something, the time steps used are variable within the models, making it difficult to clearly compare the errors associated with temporal discretization and clearly separate them from errors associated with spatial discretization.

It is indeed true that time steps vary between the TVD-FVM and the implicit method on the one hand and the implicit method without a control on the time step on the other. The latter was done on purpose to illustrate how the main advantage of an implicit scheme, i.e. being stable at time steps exceeding the CFL criterion, is counterbalanced by numerical smearing once the CFL criterion is exceeded. If we only compared simulations where the time step obeys the CFL criterion, it would make no sense to use the implicit scheme as the explicit FDM would be as fast or faster (due to the possibility of vectorization).

Before I discuss this issue further, I think it is important to note that LEMs, like solutions to any other PDE or set of PDEs, should converge as the pixel size goes to zero, or at least be relatively insensitive to the grid resolution over the range of resolutions to which the model is applied. Without this, there is no unique solution for a given set of parameter values, making it impossible to know, in the absence of an analytic solution, if one has achieved the correct solution or to objectively compare results obtained with different schemes (the focus of this paper).

We completely follow the argumentation that numerical models should converge at small resolutions. We applied an analytical solution, which per definition gives the 'the true solution' to illustrate that the different numerical methods applied in our paper indeed converge at small resolutions. Our approach to prove this is further clarified in detail under point 4.

1

50

10

25

30

35

3. Moreover, if a LEM is grid-resolution dependent then the same numerical model operating at different resolutions has to be separately calibrated to data, rendering parameter values such as D and K that should be solely functions of natural processes and material properties also functions of grid resolution. Pelletier, Geomorphology, (2010) has provided some guidance on how to make coupled hillslope-fluvial LEMs grid-resolution independent. His approach involves reframing the stream power as unit stream power (following all sediment trans- port formulae ever proposed, which is not a trivial rescaling since the contributing area generally scales with the pixel size on planar hillslopes but is relatively independent of the pixel size in convergent portions of the landscape) and modifying the strength of the diffusion term to account for the fact that changes in cross-sectional slope at valley bottoms occur over a distance equal to the valley bottom width (a property of nature), not the pixel size (not a property of nature). The random component of the model used by Campforts et al. poses a special challenge to achieving grid-resolution independence. However, one can maintain grid-resolution independence in a model with spatial random variability by generating random field(s) sampled at a resolution that represents the largest resolution the model will be applied to, then bilinearly interpolating these fields for use in versions of the model run at higher resolution. I am not suggesting that the authors adopt all (or any) of these suggestions, but I do suggest that this issue needs to be addressed in some way. The error calculation (equation (22)) simply assumes that the solution with TVD-FVM is exactly correct and any difference from this solution is an error. Without establishing grid-resolution independence it is really impossible to tell whether outputs such as Figures 8A and 8B are even unique solutions for a given set of parameter values, much less which one is more accurate.

Again, we agree with the reviewer that there is a need for a grid resolution independent solution in order to verify and compare the robustness of the different numerical schemes applied in TTLEM. We also appreciate the elegant suggestion to obtain grid independency as proposed in Pelletier 2010 and have modified the discussion of the manuscript to highlight the influence of grid resolutions. The implementation of the proposed methodology to make a numerical model grid resolution independent is however beyond the scope of our paper where we mainly want to illustrate the importance of numerical diffusion when using most frequently applied first order FDM to solve the SPL. The second message we want to bring with this paper is the suitability of a 2D variant of the TVD-FVM to simulate tectonic shortening. Although grid resolution dependency could most surely be investigated in a future release of TTLEM, we follow referee 2 in trying to present our main messages as clear as possible without drawing too much attention to the technicalities of the numerical model. For similar reasons, we decided to remove the part on grid symmetry from the manuscript and no longer discuss the different hillslope diffusion schemes implemented in TTLEM.

4. The different methods are only evaluated for a small number of cases (two grid resolutions and cases with and without a maximum time step). Error in a first order method will decrease linearly 90 as you decrease dx and it will decrease with  $(dx)^2$  for a second order method. In moving from a grid with dx=500m to dx=100m, there is a large difference in the computed values of E depending on whether or not a first-order or higher-order method is used. This is expected, but this doesn't indicate a fundamental problem with any of the numerical methods. The error associated with each of the methods is dependent on the grid resolution. So, it is a given that there will be some 95 range of grid resolutions where the differences between a 1st order and  $2^{nd}$  order method appear unacceptable (i.e. numerical diffusion is excessive relative to the prescribed diffusivity). However, what really matters in judging method accuracy is the computational time required to reach a given level of accuracy relative to an exact/converged solution. What would be most helpful is to demonstrate that TVD-FVM saves considerable computational time by providing an acceptable 100 solution at a much higher grid resolution and/or is robust for a much wider range of grid resolutions than first order methods. I suggest the following: First, for one method, perform the simulation for a range of grid resolutions (400m, 200m, 100m, 50m, 25m, 12.5m) until the solution converges, i.e. becomes essentially grid-resolution independent. Use a time step that is small

2

55

60

65

70

75

80

enough so that the solution does not depend on the time step (this probably means using a time step that yields a very low Courant number for the coarser grids, but the magnitude of the time step is likely to be similar to the magnitude of the time step needed to keep the model stable on finer grids). Then, it is easy to argue that most of the error introduced into the solution is associated with the spatial component of the problem. Second, repeat step 1 for each of the numerical methods. Assuming all simulations are run on the same machine, keep track of the time required to perform the simulations. This would allow for a more robust comparison of the different methods and would give readers a better idea of the true differences between the methods. For instance, the TVD method should converge to an grid-resolution-independent solution more quickly than the lower order methods. But how much faster? How does this depend on uplift rate or other commonly varied parameters? What are the practical implications in terms of computing time? This would give readers more guidance on the necessity of using one method over the other.

We consider this remark as very essential and would like to thank the reviewer for his suggestion on developing a grid independent 'true' solution for the SPL and TTLEM in general. We decided
 that such an approach is indeed most essential and would offer the reader much more guidance in the performance of the algorithms and provides a robust method to compare the different numerical schemes. Moreover, also reviewer 2 requested a robust framework to illustrate the performance of the numerical schemes. However, carrying out the analysis as suggested by the reviewer introduced some complexities and uncertainties which are summarized below.
 Therefore, we performed an alternative test, also covering a wide range of resolutions and we compared our numerical solution with an analytical one so that resolution effects could be analyzed.

Complications which arise when performing the analysis as outlined above mainly come down to the fact that comparing model runs with similar parameter values at different resolutions is a very tricky business. First, interpolation from the 'starting initial image' to the other resolutions (e.g. from 10 m to 400 m) will change the initial location of the drainage network to a certain extent, depending on the interpolation method used. Hence, catchments and rivers might shift in location which complicates comparison between results. Second, and this one seemed to be very important while doing the exercise, changing the resolution from e.g. 400 to 10 m results in much more possible river paths. This is illustrated in the figure below where it is shown that river distance in higher resolution images might be much longer and can take many different shapes compared to the main resolution (where river length is 400 m or 400 m × sqrt(2)).

For these reasons, when comparing models, executed at different resolutions, one is rather evaluating the effect of raster resolution and the way it is reflected in topography than comparing the performance of numerical schemes. Although the latter is of utmost importance and has been elegantly illustrated in literature (Pelletier, 2010), this is not what is required to evaluate the performance of a numerical scheme.

In order to overcome these problems, we developed the following strategy to evaluate both the computational performance and accuracy of the numerical methods:

• We only consider river cells to quantify the performance of the different numerical schemes. These rivers cells set the base level for the hillslope cells and the way these hillslope cells respond to differences in numerical schemes is illustrated by the erosion rates calculated over several catchments and illustrated in the current figure 7 of the manuscript. We agree however, that our previous approach to document the difference between the TVD scheme and the implicit schemes using a RMSE is misleading. We will therefore no longer refer to the term RMSE to document the difference between two

3

140

145

|    | numerical schemes but simply report the difference between the schemes as an offset. E.g.  |
|----|--------------------------------------------------------------------------------------------|
| 55 | the O TVD-imp represents the offset between the TVD-FVM and the implicit FDM.   |
|    | • To document real RMSE values as a consequence of numerical diffusion we performed        |
|    | the following analysis:                                                                    |
|    | 1. We initiate the analysis from the standard DEM, also used to calculate differences      |
|    | in erosion rates plotted in the current figure 7-9.                                        |
| 60 | 2. All river heads with a contributing drainage area exceeding a threshold value are       |
|    | selected (in our case $10^6 \text{ m}^2$ )                                                 |
|    | 3. The drainage network connecting these river heads with the outlet of the catchment      |
|    | is calculated. Very short river profiles

Figure .

185

190

195

5

10. The previous steps are repeated for a range of resolutions going from 950 m to 6.25 m. For each model run, the CPU time required to perform the analysis is stored.

11. Given that we have an analytical solution for all the cells of the drainage network, the numerical accuracy of the methods can be evaluated by calculating the RMSE between the three numerical methods and this analytical solution. The result of this exercise is plotted in Figure which is in fact reporting the data required by the reviewer.

We will discuss these findings in detail in the revised manuscript but note that from this analysis, one can see that it would take for example 12 times longer to obtain the accuracy of the river processes obtained with a TVD-FVM at 500 m (RMSE = 18.17, 2.89 sec) with an implicit method (cfl

200 Figure 1: DEM of standard run used in the current version of the paper to calculate catchment wide erosion rates and here used as an initial DEM to run the performance analysis outlined in the comments of the reply. The grey lines indicate the drainage network for which the solution has been calculated analytically. The blue line indicates the river profile for which model results at different resolutions are plotted in figure 2.

Figure 2: Comparison between different modelled resolutions for the river profile indicated in blue in figure 1. The green line is the 'true' analytical solution, obtained with the slope patch method of Royden and Perron (2013). The solid blue line presents the implicit solution when the CFL

Figure 3: a. Performance of the different numerical schemes calculated with the RMSE between the analytical and numerical methods. b. CPU time required to perform the model runs at the indicated resolutions.

In the discussion the authors imply that their method is really the only acceptable method for the stream-power component of LEMs. Techniques that are widely used to prevent artificial numerical diffusion in many fields of science, including MPDATA and semi-Lagrangian techniques, are implied to be inferior or less robust with no evidence. For example, semi-Lagrangian methods are deemed to be potentially of higher accuracy, but then simply dismissed as inferior to TVD-FVM because "simulation of horizontal topographic shortening would require large amounts of incremental markers to prevent numerical diffusion when interpolating the solution." This sentence confuses two different methods (semi-Lagrangian and particle-in-cell methods are not the same) and is not based on any evidence. I don't see any point in discouraging the community from trying alternative methods until they are clearly tested and shown to be inferior for a wide range of potential applications.

We do accept that our considerations were worded somewhat too strongly. We have therefore adjusted this in the new version of the manuscript. That being said, and without the intention to discourage the community from testing other numerical methods, we are confident in stating that the TVD-FVM is a relatively easy to implement numerical solution which does minimize the amount of numerical smearing

- in the solution. I did implement an adapted version of the MPDATA scheme which ultimately leads to a similar performance compared to the TVD-DVM but only after applying the limiters as pointed out in the manuscript. That makes the scheme heavier and more complex compared to the TVD-FVM and so we concluded that in this particular case, there is no need for using an MPDATA scheme. Regarding the Lagrangian schemes, we agree with the referee that the current text was confusing and we have rewritten
   the paragraph as follows:
- The numerical methods discussed so far are solved on an Eulerian grid. Eulerian grids represent immobile observations points, for which the solution of the variable, in our case topography, is calculated through time. Alternatively, Lagrangian points such as markers or particles are directly connected to the variable (topography) and evolve together with the variable over time (Gerya, 2010). An approach that
- 240 has previously been shown to be successful in preventing numerical diffusion is the Marker In Cell method. Here, the solution of the system is simulated by interpolating independently propagating Lagrangian advection markers to fixed Eulerian grid points during each time step of the simulation (Harlow and Welch, 1965). In a 1D configuration, this method would produce very accurate results when applied to solve an advection equation such as the SPL. However, simulation of horizontal topographic shortening would require large amounts of incremental markers to prevent numerical diffusion when
- 245 shortening would require large amounts of incremental markers to prevent numerical diffusion when interpolating the solution to the Eulerian grid (Gerya, 2010).

7

Some of the weaknesses of the tested numerical solutions can be reduced by LEMs that rely on irregular grid geometries. Irregular grids do, for example, allow to simulate tectonic shortening using a fully Lagrangian approach where grid nodes are advected with the tectonically imposed velocity field (e.g. Herman and Braun, 2006). ...

Minor issues:

250

1) The variable x is used for two different things (in eqn. (1) it represents one of the cardinal horizontal directions but in eqn. (2) is represents the along-channel distance).

255 We will fix this in the revised version of the manuscript.

2) There is some repetition and inconsistency in the equations. For example, there are 6 different equations for one variable (dz/dt). It would be better to use a notation that differentiates among different aspects of dz/dt (tectonic advection versus diffusive erosion/aggradation versus stream-power-driven erosion) and make it clear that dz/dt is the sum of these different components. As written, equations (1) and (6) and (9)

260 make it clear that dz/dt is the sum of these different components. As written, equations (1) and (6) and (9) are repetitive and incompatible, because they are almost the same equation, yet the left hand side of all of the equations is the same while the right hand side includes uplift in one of the equations but not in the other two.

We agree that our notation is currently not fully consistent and follow the suggestion of the reviewer to use different notations for the different sub components of the solution (eg. Eq. 6 and 9)

3) It would be helpful for the authors to address whether the method could be applied to the nonlinear stream power law (n not equal to 1), spatially variable K (e.g., strong over weak layers in sedimentary or metamorphic rocks), transport-limited fluvial processes, landscapes with a finite soil layer over bedrock or intact regolith, and other common LEM variants.

We thank the reviewer for these suggestions. For the moment the model supports (i) non-linear river incision (n~=1), variable K values, different precipitation input. Transport limited fluvial processes as well as a bedrock/regolith interface are currently not supported but are planned to incorporate in future versions of TTLEM.

4) The paper is comprehensively referenced, which I appreciate, but some of the references do not support the points being made. To take one example, McGuire and Pelletier (2016) is used to defend the use of a detachment-limited model on the basis that unconsolidated sediment can be easily evacuated from the fluvial network. This is simply untrue. Unconsolidated sediments obviously do get stored in fluvial

280 systems. Whether a detachment-limited model is a reasonable approximation depends on the application (including details such as mean grain size), and I don't think a paper that deals with small channels forming on alluvial terraces is an appropriate basis for defending the use of a detachment limited model in an LEM designed to model the large-scale evolution of mountain belts.

We agree with RC1. We will change the referencing and wording in this sentence.

285

270

5) The structure of the paper is good but the sections/subsections could be slightly improved. For example, the issue of artificial symmetry that can arise with rectangular grids is first introduced on line 206 with no prior mention or subsection break. I think this issue should be addressed in its own subsection of section 3 (as it is in section 4.2).

290 We will no longer discuss the issue of artificial symmetry in this paper as suggested by referee 2.

6) The stream power model is introduced using its nonlinear form (the exponent n is general) but the remainder of the paper, including the CFL condition (eqn. (19)), applies only to the linear case.

All the simulations could be easily performed for non-linear cases. However, we preferred linear examples when demonstrating the impact of numerical smearing on the results to enhance clarity in general. How non-linear slope dependency affects river incision is discussed in Campforts and Govers (2015) in due detail, including the way in which the CFL criterion should be adapted.

300

7) The use of D8 routing seems unsubstantiated. Dinfinity is the choice of nearly every modern LEM, because it more faithfully represents flow on hillslopes.

Dinf (or  $D\infty$ ) is certainly the flow routing scheme of choice to represent flow on hillslopes. However, in TTLEM fluvial erosion is limited to the channelized domain of the landscape and thus the flow routing scheme on hillslopes of minor significance. Nevertheless, even in the channelized domain Dinf has advantages over D8 since it enables diverging flows on landforms such as alluvial fans and braidplains. The current implementation of TTLEM, however, focuses on the modelling of detachment-limited systems or bedrock rivers where divergent flows are usually confined by valley walls. This is also consistent with other models such as Fastscape (Braun and Willett, 2013) and DAC (Goren et al., 2014) models that use the D8 flow routing scheme. We thus disagree that Dinf is the choice of the majority of

310 models that use the D8 flow routing scheme. We thus disagree that Dinf is the choice of the majority of modern LEMs. Still, we like to stress that we do not exclude to implement Dinf or other multiple flow direction algorithms in a future version of TTLEM, in particular since the topological sorting algorithm (Braun and Willett, 2013; Heckmann et al., 2015) is equally suitable for the efficient computation of flows on thus derived networks.

315

305

8) Please use lat/lon or UTM coordinates in Fig. 2. If these are UTM coordinates, please specify. We will fix this in the updated version of the manuscript.

9) The method of the paper is referred to as TVD-TVM throughout the abstract but TVD-FVM in the
 paper. If this is not a typo, please explain the difference between these abbreviations.
 We will fix this in the updated version of the manuscript.

10) w\_A and w\_k are introduced in the equation but then (unless I missed it) never discussed again (not even in the table of parameter values).

- 325 These parameters are weighting parameters used to scale for changes in precipitation and lithology. We will clarify this.

Earth Surf. Dynam. Discuss., 330 doi:10.5194/esurf-2016-39-RC2, 2016 © Author(s) 2016. CC-BY 3.0 License. Interactive comment on "Accurate simulation of transient landscape evolution by eliminating numerical diffusion: the TTLEM 1.0 model" by Benjamin Campforts et al.

335

**Anonymous Referee #2**

Received and published: 17 August 2016

- We thank referee 2 for her/his comments, which helped us to improve the quality of the manuscript. Our 340 replies are in blue. Throughout this reply, we will also refer to the answers formulated in the author comments on referee 1 (further referred to as RC1) where we also added some figures for clarification.
- Campforts et al. addresses an important problem for fluvial landscape evolution models: numerical 345 diffusion of the solution to the stream power advection equation. The authors first of all present a solution to the problem based on a higher-order flux-limiting method (TVD-TVM), and secondly, they outline a new modeling platform (TTLEM), which makes use of TVD-TVM and is available to everyone as part of the TopoToolbox.
- Overall, my opinion is that numerical accuracy of fluvial landscape evolution models has received too little attention in the past, and it is therefore good to see the authors address it here. The method proposed 350 to reduce numerical diffusion is convincing, and the damping of numerical diffusion in stream-power advection as well as in tracking horizontal tectonic displacements is significant. I hope that this contribution gets published in Esurf, although I do have some concerns and suggestions, which I list below:
- We are grateful for RC2's appreciation of our work. We also appreciate the constructive comments which 355 will help us to enhance the overall quality and readability of the manuscript.

**General comments:**

- First of all, I think the dual purpose of the manuscript: 1) discussing numerical diffusion and presenting 360 TVD-TVM, and 2) presenting TTLEM as a more general landscape evolution model leads to a rather diffuse and ackward structure of the text. The main strength of this text is in my opinion the focus on numerical diffusion and the presentation of TVD-TVM, but the TTLEM presentation calls for many details that are not needed to address this issue (see for example Fig. 1). For example, because the introduction focuses mostly on the influence of numerical diffusion, it is hard to understand the motivation
- for the first couple of experiments focusing on drainage networks and the influence of different hillslope 365 models. I would strongly recommend simplifying the flow of the manuscript focusing more exclusively on the issue of numerical diffusion. Likewise the authors should consider skipping the first two experiments and in stead perform more like the one shown in Fig. 7. I think that it would increase the impact of the contribution, and the presentation of TTLEM could perhaps be saved for another manuscript in a more software-oriented journal. 370
- We follow the advice of the reviewer to focus the entire manuscript on the role of numerical diffusion in landscape evolution modelling. We will therefore remove the two first experiments (e.g. the role of hillslope diffusion and the presence of artificial symmetry) from the paper. Nonetheless, we consider this paper as the first description of the new TTLEM simulation software. Therefore, we will move the flow
- chart illustrating the different modules of the model to the appendix of the paper along with the picture 375 illustrating the functionality of the different hillslope response schemes. We consider TTLEM as a tool for the community which can be used to reconstruct landscape evolution as well as to test hypotheses. The latter might require a combination of insights in the different existing modules as well as a guidance on how to add new modules. We feel that both objectives, require an overview of the software in its 380 present shape.

Secondly, I suggest the authors give a short introduction to basic knowledge about numerical diffusion in advection problems. This could be inspired by simple textbook material and use linear advection as a

starting point. By this the authors could avoid some awkward reflections, like in line 378: it is not at all
 counterintuitive that time steps smaller than the CFL criterion leads to more numerical diffusion. Most
 numerical analysis textbooks I know of give very simple explanations for why numerical diffusion is
 minimized exactly at the CFL criterion. Overall, I think the authors can make better use of basic textbook
 wisdom to prepare the reader for the main points of the manuscript.

In the revised manuscript, the readers are introduced into the issue of numerical diffusion when solving hyperbolic partial differential equations (section 3.2). We also updated some references to excellent textbooks on this matter (Harten, 1983; Toro, 2009). An extended discussion on numerical diffusion when solving the stream power law can be found in Campforts and Govers (2015). We will also rephrase the sentence in line 378 although we find it important to document these findings which are indeed well discussed in numerical textbooks but less well known/introduced in the earth surface community.

395

400

405

425

430

435

Finally, while I fully appreciate the comparison experiments between the different numerical methods, I suspect that it is not completely fair.

Part of this answer is addressed in the reply to RC1 where we illustrate how the analytical slope patch method (Royden and Taylor Perron, 2013) is used to evaluate the performance of the different numerical schemes.

The main advantage of the implicit method (as FastScape by Braun and Willett) is that it becomes more compute efficient at high spatial resolution than the explicit methods, simply because it is not similarly constrained by the CFL condition. Thus, if explicit and implicit methods were compared in experiments with similar compute time (which I think they should be), would the implicit method not allow for finer spatial resolution than the explicit method? If so, would the finer spatial resolution in combination with the larger time steps not reduce the numerical diffusion of the implicit method? I am not questioning the advantages of TVD-TVM here. I just feel that the authors are not appreciating the real strength of the

410 This is an interesting remark that we address in a revised version of the manuscript. We hope that the additional analysis outlined in our comment to RC1 will provide more insight into the trade-offs between numerical accuracy and computational efficiency. The answer to the referee's question comes in multiple points.

implicit method, which is how the compute time scales with spatial resolution.

- An essential characteristic of an implicit scheme like that of Braun and Willett is that it fails to allow for 'vectorization' which is in contrast to explicit methods (like TVD). By vectorization, we mean ways to exploit single-instruction multiple-data parallelism. Hence, the fact that TVD requires more operations per execution and requires a time step which obeys the CFL criterion may partly compensate for sequential looping through all stream network nodes required by the implicit scheme. From the analysis presented in our discussion of the comments of RC1, we show that both schemes end up running in almost the same time. We will address this point in the new version of the manuscript.
  - It is important to note that rivers only occupy part of the landscape. Although TTLEM indeed allows to simulate all cells as rivers cells (as suggested in comment on line 206), we do not test this configuration as we consider it of little use in real world landscape evolution where hillslope processes may dominate where drainage area drops below a threshold value. Hence, while refining the resolution does indeed result in more accurately simulated river elevations, the computational overhead related to hillslopes processes which comes with refining the grid resolution is unacceptably large at the spatial scales and resolutions that we consider. Also notice that even at very high spatial resolutions (6.25 m), the TVD method is still more accurate compared to the implicit (cfl<1) method.
  - We appreciate the remark of the reviewer that the higher spatial resolution, which is in principle allowed by the implicit method for similar timescales, is the real strength of the implicit method. This argument is exactly the reason why we simulated the landscape using both an implicit method which is free of any time criterion (and where dt is set by the main model time step, e.g. 2e4 yr) and one simulation where a CFL is applied to the implicit method. The latter was done on purpose to illustrate how the main advantage of an implicit scheme,

i.e. being stable at time steps exceeding the CFL criterion, is counterbalanced by numerical smearing once the CFL criterion is exceeded. If we only compared simulations where the time step obeys the CFL criterion, it would make no sense to use the implicit scheme as the explicit FDM would be as fast or even faster (due to the possibility of vectorization). Furthermore, it 440 is not only the inherent nature of an implicit scheme which is not suited to properly simulate propagating knickpoints. If very large timescales are applied in landscape evolution models, uplift is inserted very suddenly at the beginning of the time step. This results in unrealistic simulations where uplift is a discrete stepwise function rather than a continuous function (e.g. the sine waves used in this paper). In Fig. 2 of this file, we have shown two extremes, i.e. a 445 configuration where CFL<1 and one where CFL >>1. One could argue that intermediate solutions (e.g with CFL closer to 1) would result in more desirable results than the one shown with the dotted lines in Fig. 1-3 of RC1. This is true but, given that computational gains are marginal and numerical accuracy will never be higher than the implicit method simulated at CFL< 1 (solid blue lines), we see little reason to follow such an approach when simulating 450 transient landscape evolution.

• To summarize, a first order implicit scheme is not suited to properly simulate propagating knickpoints in detachment limited erosional basins. First order implicit methods are therefore only suited to simulate configurations where transiency, caused by local base level falls, tectonic faults or lithological contacts can be considered to be minor.

More specific comments: 460 Line 30: "availability of potential energy" Line 85: delete "most" Eqn 1: Why are vx and vy bold? Because they are representing velocity fields being variable in space. Eqn 2: Are wk and wa used for anything here? If not flush them out. 465 They are used as weighing factors to introduce the impact of variable lithological strength an precipitation in the model. We will further clarify this in the updated manuscript. We now refer to them as well in the discussion section. Line 102: what is "eroding settings"? Where the detachment limited assumption holds. 470 Eqn 3: The diverge operator should include a dot between nabla and qs OK Line 113: hillslope erosivity and erodibility. What is the difference? Should be simple erodibility. Erosivity can be removed Eqn 7: Again, is the variability on m really needed to demonstrate the points of numerical diffusion? If 475 not skip it to clean the text. More complicated means less convincing. Point taken. Section will be removed in the updated manuscript. Eqn 8: I do not understand the effect of densities here. Is U not simply uplift of the surface? If so, I guess the densities should be on the second term, right? 480 The way it was written in the previous version of the manuscript was actually correct. The correction for

the bedrock versus soil bulk densities is required on hillslopes where erosion and deposition in governed by soil fluxes (Perron, 2011). Nonetheless, we agree that the way in which this was presented was somewhat confusing and we adapted the presentation of the mass balance equation in the new version of the manuscript.

485 Line 153: "...transforms returns..." Eqns 11-17: The use of subscripts seems inconsistent. We only solve one component of the differential equation. The full deriviation can be found in the textbook we refer to or online in are GitHub Code. Line 192: "... is similar than the one..."

490 Eqn: 19: I guess A varies by several orders of magnitude in the grid. Please discuss

12

the CFL criterion in the light of this. Is max(A) used here?

Fixed

495

Line 199: Description of the inner time step is confusing, and I do not understand why it is needed. Again I suspect that it is the general presentation of TTLEM that stands in the way for a clear and concise presentation of the numerical experiments.

We will clarify this further in a revised version of the manuscript. An inner time step is needed because hillslope processes which are diffusive in nature allow the use of semi-implicit methods used to solve them. Here, the implicit nature of the schemes can be fully exploited and large time steps can be used to solve the equations (Perron, 2011). The TVD method which is explicit, on the other hand does not allow such big time steps and does require the main model time step to be split up in so called 'inner time steps'.

Lines 206-215: This kind of randomness should be avoided here. The authors are documenting the level of numerical diffusion in different numerical techniques, and in this process it is very important that we know what advection equation is solved. m seems to be varied in order to make the drainage networks look more realistic. But that is not important here. And by the way: varying m randomly does not remove

- 505 look more realistic. But that is not important here. And by the way: varying m randomly does not remove the grid dependency (which is inherent to stream-power advection and D8 drainage), it just obscures the close links between the grid, the (random) variability of m, and the drainage network. Please keep m fixed and the equations as simple as possible! Section removed
- 510 Section 3.4 is not well written. In spite of carefully reading the text I am still confused about how hillslope
- processes are implemented. But more importantly: Can the experiments documenting numerical diffusion not be run without hillslope processes? This would require that Ac=0 in Eqn 8, but why not? It seems a bit silly to deliberately add physical diffusion to an experiment were one wants to measure numerical diffusion? The authors should consider if the experiments can be made simpler (see first general comment above). Skipping hillslope processes and deleting this section could be a quick fix.
- As outlined above, we agree with the reviewer that the experiments on hillslope diffusion and varying values for m are distracting for the main message of the paper. We will also further motivate our choice for the D8 algorithm in the updated manuscript (see also RC1). However, for reasons also discussed above, we did not remove the hillslope processes from our model to explicitly address how numerical diffusion in channel incision affects hillslope diffusion and ultimately basin wide erosion rates.
- Section 4: I recommend skipping the first two experiments on hillslope processes and drainage networks (or save them for another paper). This would free up space to dig deeper into advection and numerical diffusion.

Fixed, section removed from the manuscript

- 525 Line 276: I am not impressed by this strategy. I agree that the artificial symmetry is a problem, but at least we know where it comes from. Fixing this by introducing variability in the exponent m obscures the link between model input and model output, which is otherwise critical for use of computational experiments. Variability on K is better, because the linear scaling does not alter the form of the equation. Fixed, section removed from the manuscript
- Line 344: So, what happens if the grid resolution is lowered to 10 m?
   See RC 1
   Line 391: overcomes -> reduces

Fixed

Line 403: A small time step is not the essential factor here. The implicit method first of all offers a fine spatial resolution in combination with a large time step. The advantage of this combination should be explored more.

This issue is discussed in the reply to the major comments above.

Line 464-474: All of this seems rather irrelevant to the main points of this study. See first general comment.

540 We will consider moving part of the paragraph to the appendix in the revised version of the manuscript. Line 481: "... the current debate. ..." calls for references. Fixed

Fig. 1: I almost get dizzy by looking at this. What is the point of showing this level of complexity in the first figure?

545 We will skip this figure and add it to the appendix

Fig. 2: While this is interesting I do not understand the motivation. The introduction spins me up to read about numerical diffusion, not this.

We will skip this figure and add it to the appendix

Fig. 3: Same comments as for Fig. 2.

550 We will skip this figure

Fig. 4: This is a nice, simple figure and to me the extension of this existing result to 2D simulations is the essential contribution of this study. This figure could be a great opening figure.

Point taken

Fig. 7: If the authors choose to follow my advice and skip the first experiments, then more like this could
be performed. It would be useful to see experiments with different setting of m and n (linear vs. nonlinear).
Also to have experiments at finer spatial resolution where the advantages of the implicit method should
start to kick in.

See discussion above and figures in RC1. We will remove the first three figures from the manuscript.

Fig. 9: It is good to see the difference between methods here, but it would also be great to see pictures of the two separate erosion rates. I wonder if knickpoints can be recognized in both?

- Fig. 9 illustrates the difference between erosion rates for the two numerical methods. In our opinion the addition of another figure showing the erosion rates for each method is not very meaningful as the differences in erosion patterns and rates would be less clear. With respect to the knickpoints it is important to consider that the use of a different numerical method does not change the average speed of knickpoint advection (see Campforts and Govers, 2015), but it does strongly affect the evolution of the gradient of
- the knickpoint: we will add this clarification in the revised version of the manuscript. Hence, it is not meaningful to compare maps of knickpoint locations. Fig. 10: great figure

Thanks

[revised manuscript text omitted]

(1)

(2)

where  $\mathbf{v}_{\mathbf{x}} \mathbf{\underline{u}}$  and  $\mathbf{v}_{\mathbf{x}} \mathbf{\underline{v}} [\mathbf{L} \mathbf{T}^{-1}]$  are the tectonic displacement velocities in the cardinal  $\mathbf{x}$  and  $\mathbf{y}$  directions (horizontal  $\mathbf{\underline{\mu}}$  and vertical  $\mathbf{\underline{v}}$ ), respectively.

**700 2.2.River incision**

Detachment limited fluvial erosion  $(\partial z/\partial t)_{fluv}$  is calculated based on the well knownestablished relation between the channel gradient and the contributing drainage area (*A*), also referred to as the Streamwith the -stream Power power Law-law (SPL) (Howard and Kerby, 1983):

$$\left(\frac{\partial z}{\partial t}\right)_{fluv} = -K(A)^m \left(\frac{\partial z}{\partial x_{\Gamma}}\right)^n$$

705

The equation is solved on a dendritic stream network domain  $\Gamma$  where  $x_{T_k}$  refers to the distance from the outlet.  $A_k[L_{cl}^2]$  is catchment area and K [ $L^{1-2m}$  t-1] is an erodibility parameter that depends on local climate, hydraulic roughness, lithology and sediment load. *K* can be adapted to reflect local variations in erodibility by using a scaling coefficient  $w_k$  [dimensionless]. In ease of uniform erodibility,  $w_k$  is set to one. *A* is the drainage area, which is used as a proxy for the local discharge. Similar to *K*, *A* can be corrected for regional precipitation variabilities through a scaling coefficient  $w_k$  [dimensionless].

*m* and *n* represent-are the area and slope exponents: their values reflect hydrological conditions, channel width, as well as the
 dominant erosion mechanism. *K*, *m* and *n* are interdependent and it is usually impractical to constrain any of their values alone
 (Croissant and Braun, 2014; Lague, 2014). Thus, many studies provide estimates for the *m/n* ratio. For *m/n* ratios between 0.35

| Formatte | ed: Subscript              |
|----------|----------------------------|
| Formatte | ed: Subscript              |
| Formatte | ed: Caption                |
|          |                            |
| Formatte | ed: Font: Bold, Not Italic |
| Formatte | ed: Font: Bold, Not Italic |
| Formatte | ed: Font: Bold             |
| Formatte | ed: Font: Bold             |
| Formatte | ed: No Spacing             |
|          |                            |

|  | Formatted: Caption                                              |
|--|-----------------------------------------------------------------|
|  | Formatted: Font: 9 pt                                           |
|  | Formatted: Space After: 12 pt                                   |
|  | Formatted: Font: Not Italic, Font color: Text 1                 |
|  | Formatted: Font: Not Italic, Font color: Text 1                 |
|  | Formatted: Font: Not Italic, Font color: Text 1, Subscript      |
|  | Formatted: Font: Not Italic, Font color: Text 1                 |
|  | Formatted: Font: Not Italic, Font color: Text 1                 |
|  | Formatted: Font: Not Italic, Font color: Text 1,
Superscript |
|  | Formatted: Font: Not Italic, Font color: Text 1                 |
|  | Formatted: Font: Not Italic, Font color: Text 1                 |

and 0.8, *K* values span several orders of magnitude between  $10^{-10} - 10^{-3} \text{ m}^{(1-2m)} \text{ yr}^{-1}$  (Kirby and Whipple, 2001; Seidl and Dietrich, 1992; Stock and Montgomery, 1999). In order to represent fluvial sediment transport, it has previously been proposed to add a diffusion component (Rosenbloom and Anderson, 1994). However, we follow others in assuming that in eroding settings, detachment limited erosion is controlling landscape evolution and is represented by the advection equation represented in Eq. (2) (Attal et al., 2008; Goren et al., 2014; Howard and Kerby, 1983; Whipple and Tucker, 1999).

**2.3. Hillslope processes**

715

720

River incision drives the development of erosional landscapes by changing the base level for hillslope processes. Steepening of hillslopes subsequently leads to increased sediment fluxes from hillslopes to the river system. Hillslope erosion-denudation  $(\partial z/\partial t)_{hill}$  is equal to the divergence of the flux of soil/regolith material (**q**s, [L3 L-1 T-1]):

$$\left(\frac{\partial z}{\partial t}\right)_{hill} = -\nabla \cdot \mathbf{q}_s$$

Different geomorphological laws describe hillslope response to lowering base levels. The model of linear diffusion assumes that the soil/regolith flux is proportional to the hillslope gradient (Culling, 1963):

$$\mathbf{q}_{s} = -D\nabla z$$

- 725 where D is the diffusivity [L2t-1] that parameterizes hillslope erosivity and erodibility and determines rate of soil/regolith creep. Linear hillslope diffusion produces convex upward slopes. Main controls on variations of D include substrate, lithology, soil depth, climate and biological activity, amongst others. Values of D vary widely and range between 10-3 and 10-1 m2 yr-1 for slopes under natural land use (Campforts et al., 2016; DiBiase and Whipple, 2011; Jungers et al., 2009; Roering et al., 1999; West et al., 2013, Linear hillslope diffusion produces convex upward slopes.
- Field evidence37 however, suggests that this the linear formdiffusion model is only rarely appropriate (Dietrich et al., 2013). Instead, hillslopes often tend to have convex-planar profiles because rapid, ballistic particle transport and shallow landsliding dominate as soon as slopes approach or exceed a critical angle (DiBiase et al., 2010; Larsen and Montgomery, 2012). To account for this rapid increase of flux rates with increasing slopes, Andrews and Bucknam (1987) and Roering et al. (1999) proposed a nonlinear formulation of diffusive hillslope transport, assuming that flux rates increase to infinity if slope values approach a critical slope *S*c:

$$\mathbf{q}_{s} = -\frac{D\,\nabla z}{1 - \left(|\nabla z|/S_{c}\right)^{2}}$$

Main controls on variations of *D* include substrate, lithology, soil depth, climate and biological activity, amongst others. Values of *D* vary widely and range between 10-2 and 10-4-m2 yr4 for slopes under natural land use (Campforts et al., 2016; DiBiase and Whipple, 2011; Jungers et al., 2009; Roering et al., 1999; West et al., 2013).

**740 2.4. Overall landscape evolution**

In summary, TTLEM solves the following partial differential equation PDE, where by an explicit distinction is made between river and hills lope cells, based on a threshold contributing area,  $A_c$  cells sculpted by fluvial versus hills lope processes is made.:

s: First, it simulates the horizontal tectonic displacements over the entire model domain:

| + |
|---|---------------------------|
| g |                           |
| ь |                           |
| n |                           |
|   |                           |
|   |                           |
|   |                           |
|   |                           |
| + | Formatted: Caption        |
|   |                           |
|   |                           |
|   |                           |
| s |                           |
|   |                           |
|   |                           |
|   |                           |
|   |                           |
| 4 |
|   | ·                         |
|   |                           |
|   |                           |

(3)

(4)

(5)

| -                | Field Code Changed                  |
|------------------|-------------------------------------|
| (                | Formatted: English (United States)  |
| $\left( \right)$ | Formatted: English (United States)  |
|                  | Formatted: English (United Kingdom) |

| Formatted: Caption |  | FOI | mat | tea: | Capt | ion |
|--------------------|--|-----|-----|------|------|-----|
|--------------------|--|-----|-----|------|------|-----|

Field Code Changed

$$\frac{\partial z}{\partial t} = \mathbf{v}_{\mathbf{x}} \frac{\partial z}{\partial x} + \mathbf{v}_{\mathbf{y}} \frac{\partial z}{\partial y}$$

Second, TTLEM simulates detachment limited river incision for the parts of the landscape that are predominantly sculpted by fluvial processes. We determine that domain where contributing drainage area (A) exceeds a critical drainage area ( $A_e$ ):

$$\frac{\partial z}{\partial t} = U + \left( v_x \frac{\partial z}{\partial x} + v_y \frac{\partial z}{\partial y} \right) - \left( K w_K \left( w_A A \right)^{(m + \operatorname{var}(m))} \left( \frac{\partial z}{\partial x} \right)^n \right)$$
(7)

where var(m) refers to the variability on m which is explained further (Eq. (20)).

Third, we define the hillslope domain where  $A < A_{c}$ . Topographic changes in this domain are calculated by:

$$\frac{27}{2} = U - \nabla \mathbf{q}_s \tag{8}$$

where  $\rho_{\mu}$  and  $\rho_{\mu}$  are the bulk densities of the bedrock and the regolith material, respectively [m L3]. The formulation of Eq. (8) implies that we assume that hillslopes are generally covered by regolith and/or soil.

| $\partial z (\partial z)$                                             | $\int U + \left(\frac{\partial z}{\partial t}\right)_{flu}$                                                                                                      | for $A \ge A_c$ |
|-----------------------------------------------------------------------|------------------------------------------------------------------------------------------------------------------------------------------------------------------|-----------------|
| $\overline{\partial t} = \left(\overline{\partial t}\right)_{td}^{+}$ | $\begin{cases} U + \left(\frac{\partial z}{\partial t}\right)_{flut} \\ \frac{\rho_r}{\rho_s} U + \left(\frac{\partial z}{\partial t}\right)_{flut} \end{cases}$ | for $A < A_c$   |

**3.** where an explicit distinction between cells sculpted by fluvial versus hillslope processes is made. Rivers are assumed to incise directly into bedrock whereas material fluxes on slopes are assumed to mobilize -either soil or regolith, having a different bulk density than the bedrock. This is accounted for by multiplying the rock uplift rate with the density ratio between  $\rho_{z}$  and  $\rho_{z}$  [m L3] representing the bulk densities of the bedrock and the regolith material respectively -(Perron, 2011). The fluvial domain is determined by the cells having a contributing drainage area (*A*) exceeding a critical drainage area ( $A_{c}$ ).

**760 3. Implementation and numerical schemes of TTLEM**

Our main motivation to develop TTLEM is to provide users with a multi-process landscape evolution model that has a good overall computational performance and high numerical accuracy. TTLEM is predominantly–written in the MATLAB programming language; to reduce run times, however, TTLEM encompasses some C-code where this significantly improves performance (e.g. for the non-linear hillslope diffusion algorithm of Perron (2011)). Integrating TTLEM into TopoToolbox enables running the model, visualizing and analyzing its output in the same computational environment.

Figure 1 shows a schematic representation of the TTLEM workflow. Users can configure the tectonic setting by providing (i) a 2D or 3D array that represents spatially and spatio-temporally variable vertical uplift patterns, respectively, and (ii) two matrices to represent horizontal velocity fields ( $y_{xy}$  and  $y_{yy}$ ). TTLEM accepts synthetic topographies and real world DEMs and leaves users with full control on model parameter values. In the following sections, we will discuss the numerical methods involved-used in TTLEM to solve the PDEs described in section 2. The section numbers correspond to the processes indicated

involved\_used in TTLEM to solve the PDEs described in section 2. The section numbers correspond to the proces in the workflow model flowchart in the appendix (in-Fig. A1). Formatted: Stijl\_BC, Indent: Left: 0.63 cm

19

765

<del>(6)</del>

**3.1. Drainage network development**

[revised manuscript text omitted]

$$\begin{split} f^{LO}_{i+\frac{1}{2},j} &= \alpha_0 v_{i+\frac{1}{2},j} \, z^k_{i,j} + \alpha_1 v_{i+\frac{1}{2},j} \, z^k_{i+1,j} \\ f^{HI}_{i+\frac{1}{2},j} &= \beta_0 v_{i+\frac{1}{2},j} \, z^k_{i,j} + \beta_1 v_{i+\frac{1}{2},j} \, z^k_{i+1,j} \end{split}$$

The low order fluxes are solved with a first order explicit upwind Godunov scheme (1959):

$$\alpha_{0} = \frac{1}{2} (1 + sign(\mathbf{v}))_{and} \alpha_{1} = \frac{1}{2} (1 - sign(\mathbf{v}))$$

The high order fluxes are solved with a Lax-Wendroff scheme (1960):

$$\beta_0 = \frac{1}{2} \left( 1 + \mathbf{v} \frac{\Delta t}{\Delta x} \right)_{and} \beta_1 = \frac{1}{2} \left( 1 - \mathbf{v} \frac{\Delta t}{\Delta x} \right)$$

From Eq. (10), Eq. (11) and Eq. (12) it follows that:

[revised manuscript text omitted]

- 855 While the comparison of different numerical methods can provide valuable insights with respect to their relative accuracy and performance, the ultimate test is the comparison of numerical results with an analytical solution of the PDE.
  Ap aAnalytical solutions are fully correct and are evidently grid resolution independent, contrary to numerical solutions where model parameter values might depend on the grid resolution (Pelletier, 2010). However, they are not universally applicable. We implemented an analytical solution for the stream power law was implemented to test the overall model performance and
- 860 to obtain as an independent benchmark to compare the performance of the different numerical schemes implemented in TTLEM under conditions where an analytical solution can indeed be obtained. Moreover, analytical solutions are grid resolution independent, contrary to numerical solutions where model parameter values might dependent on the model resolution (Pelletier, 2010).
- In the following, the strategy to investigate the performance the model is outlined. The analysis is initiated from First wey
   created an artificial DEM where a steady state between uplift and erosion is reached. From this DEM, the drainage network and corresponding river elevations areare extracted by selecting all cells exceeding a threshold value (in our case 106 m2). Very short river profiles (<10 km) areare not retained, in the analysis to improve the performance. Subsequently, landscape evolution is simulated using the numerical models documented in the previous sections assuming spatially invariant uplift rates. At the end of the model runs, river elevations that were analytically calculated using the pre-uplift, steady state river profiles as input. Analytical solutions for the stream power law ean-beare obtained using the slope patch method of-Royden and Perron (2013). Thise method is based on a non-dimensionalisation of the stream power law. Therefore, Longitudinal river profiles areare converted to a dimensionless height (λ) and distance (γ):</li>

$$\frac{\lambda = \frac{z_x}{h_0}}{\chi = \frac{A_0^{m/n}}{h_0} \int_0^x \frac{dx}{A_x^{m/n}}}$$

23

(19)

(18)

where  $\frac{1}{26}$  represents the dimensionless elevation along the river profile,  $\frac{h_0}{h_0}$  is a reference length scale (typically set to 1 m) and A0 is a reference value for the drainage area (typically set to 1 × 106 m2). To properly integrate over abrupt changes in the drainage area along the rivers, Eq. (19) is solved using the rectangle rule (Mudd et al., 2014). Steady state river profiles (equilibrium between erosion and uplift) are represented as straight lines in this non-dimensional coordinate system. To

| /         | Formatted Table |          |
|-----------|-----------------|----------|
| /         | Formatted       |          |
| ĥ         | Formatted       |          |
| 7         | Formatted       |          |
|           | Formatted       |          |
|           | Formatted       |          |
|           | Formatted       |
     |
|           | Formatted       |          |
|           | Formatted       |          |
|           | Formatted       |
     |
|           | Formatted       |          |
| II.       | Formatted       |          |
| l         | Formatted       |          |
|           | Formatted       | (        |
|           | Formatted       |          |
|           | Formatted       |
     |
| 1         | Formatted       |          |
| 7         | Formatted       |          |
| l         | Formatted       |          |
| /         | Formatted       |          |
| Ι,        | Formatted       |          |
| /         | Formatted       |          |
| /         | Formatted       |
     |
|           | Formatted       |          |
|           | Formatted       | (        |
| -         | Formatted       |          |
| -         | Formatted       |          |
| $\langle$ | Formatted       |
[]   |
| /         | Formatted       |
     |
| /         | Formatted       |
[]   |
|           | Formatted       |
[    |
| ,         | Formatted       |          |
| ļ         | Formatted       |          |
| /         | Formatted       |          |
| /         | Formatted       |          |
| /         | Formatted       |          |
|           | Formatted       |          |
|           | Formatted       |          |
| _         | Formatted       |   |
| -         | Formatted       |          |
| 1         | Formatted       |   |
| 1         | Formatted       |
     |
| 1         | Formatted       | $\equiv$ |
| 1         | Formatted       |   |
| /         | Formatted       |          |
|           |                 |          |

890

880

properly integrate over abrupt changes in the drainage area along the rivers, Eq. (19)(19) is solved using the rectangle rule (Mudd et al., 2014). In case of a Steady state between erosion and uplift, non-dimensional river profiles are represented as a straight-line in non-dimensional coordinates. From these non-dimensional river profiles The slope patch solution can subsequently be applied for temporally variant but spatially invariant uplift rates assuming an initial elevation of the river profile. The analytical slope patch solution method developed by Royden and Perron (2013) 
[revised manuscript text omitted]
 performanceAll- all simulations are runWe use a with synthetically generated landscapes for all simulations as a starting-initial condition surfaces-because we are interested infocus on evaluating the model's performancethe evaluation of the functionality of the model and not on the correct simulation of the evolution of a particular landscape or region. Hence, our simulations are uncalibrated and results remain untested were not compared with against an actual -'true'-landscape: however, the chosen parameter values are realistic (e.g. Gasparini and Whipple, 2014; Whipple and Tucker, 1999) maybe provide one two refs for these parameter values). We distinguish between the effects on simulated river incision on the one hand and on simulated tectonic displacement on the other. To investigate the accuracy and implications of river incision methods, we compare two different explicit TVD-FVM— with the first-order implicit FDM numerical schemes and further differentiate between the and-implicit FDM\_solution where no limitation is set on the time step and the implicit solutionFDM where the CFL criterion limits the time step lengthis-limited by the CFL Formatted: No Spacing

eriterion. To investigate the accuracy and implications of river incision methods we compare an explicit first order Godunov method (GM) with the 2D TVD-FVM.

**995 4.1. River incision**

1000

1005

1010

3.5.4.1.1. 1D river incision

Ideally, The The impact of numerical diffusion on propagating river profile knickpoints can most clearly be is demonstratedmost obvious in situations where an analytical solution is available. The first simulation illustrates such a situation, with an artificial river profile characterized by a major knickzone between 8 and 12 km from the river head (Fig. 41). We assume that the drainage area is increasing in proportion to the square of the distance and uplift equals zero. For this simplifiede configuration, an analytical solution for the SPL can be found usingrelies on the method of characteristics (Luke, 1972). Notwithstanding the relatively high spatial resolution of 100 m, both implicit and explicit first order implicit Finite Difference Methods (FDMs) suffers from clear considerable numerical diffusion when river incision is calculated over a time span of 1 Myr (Fig. 41). The TVD-FVM systematically achieves a much higher accuracy, a finding that is systematic, occurring over a wide range of spatial resolutions and parameter values (Campforts and Govers, 2015).

**4.1.2. Drainage network**

The second simulationwe assess- overall numerical accuracy of the entire drainage network is assessed using spatially and temporally constant values for all model parameter values (Table 1) and assuming a fixed drainage networks (seeis assessed using the approach outlined in section 3.4). Model performance is evaluated using a simple model set-up with spatially and temporally constant values for all model parameter values and assuming fixed drainage networks. We first create a steady-state artificial landscape that we initialize with uniformly distributed random elevation values between 0 and 50 m on a 50 km values with a spatial measuring of 100 m (Maria S2). Landscape cuplution is simulated using Dirichlet boundary

× 100 km grid with a spatial resolution of 100 m (Movie S3). Landscape evolution is simulated using Dirichlet boundary conditions and by inserting spatially and temporally uniform vertical uplift of 1 km Myr-1 over a period of 150 Myr. Outer model time steps are set to 5 × 104 yr. Parameter values for river incision and hillslope response are constant in space and time and are reported in Table 1. Figure 2 shows the resulting steady state landscape.

We impose four consecutive sinusoidal uplift pulses of equal magnitude to this artificial landscape over 1 Myr. Uplift pulses have a wavelength of 0.25 Myr and an amplitude of  $3 \times 10^{-3}$  m yr-1 (Fig. 3). Theis We repeat All the simulations are performed using with three different numerical schemes to simulate river incision (implicit FDM without time step limitation, implicit FDM with time step limitation (CFL condition applied) and TVD-FVM), each at-and 22 different spatial resolutions (6.25,

- 12.5, 25, 50, 100, 150, ...950 m). Hillslopes are simulated using linear hillslope diffusion in combination with threshold slopes, a configuration typically used to simulate landscape evolution at the geological timescales (e.g. Goren et al., 2014). The threshold slope is set to a value of 0.8 m m-1 and hillslope diffusivity is set to a value of 0.01 m2y-1. ModelThe computational performance is assessed by calculating the CPU time required to perform a 1 Myr simulation. In order to facilitate the high resolution run (at 6.25 m where the spatial domain covers 7950 × 15950 cells) all model runs were executed on one computational node of the Flemish Super Cluster (VSC) using a single core (Broadwell, E5-2680v4) and featuring 128 Gi+B
- RAM. We evaluate the numerical performance of the schemes and the impact of spatial resolution against an analytical solution (slope patch method) for the entire drainage network. Independent from the numerical simulations, river evolution is calculated using the slope patch method for the entire drainage network, represented by all cells exceeding 1 km2 (indicated in grey on Fig. 2).
- 1D30 Figure 4 displays the comparison between the numerical methods and the analytical solution. Note that (The initial river profile (grey line) slightly differs depending on thespatial resolution-considered due to interpolation of the steady-state artificial landscape with a spatial resolution of 100 m. The results showse figures illustrate how thethat TVD-FVM and implicit

**Formatted: Font: 12 pt**

**Formatted: No Spacing**

1040

numerical solutions converge when model resolution is increased. In case no CFL criterion is imposed on the solution, however, Tthe implicit solution deviates from those adhering to the CFL criterion. This does not converge in case no CFL eriterion is imposed to the solution. The latter was done on purpose to illustrates that there is trade-off between how increased numerical smearingaccuracy countervails the and gain in numerical main advantagestability of for an implicit scheme at long time steps, i.e. being stable at time steps exceeding the CFL criterion, is counterbalanced by numerical smearing once the CFL eriterion is exceeded. In addition, The fact that thean implicit scheme at high spatial resolution fails to converge to an analytical solution withif time steps are large time steps is not converging at high resolutions, is however only partly explained by the first order spatial accuracy of the scheme. If very large timescales are applied in landscape evolution models, since uplift is inserted discretelyvery suddenly at the beginning of theeach time step. This results in unrealistic simulations where uplift is a discrete stepwise function rather than a continuous function (e.g. the sinoidal uplift historye waves used in this paper used here) and that inserts artificial shocks in the solution,

1045

1055

Figure 5 illustrates that the TVD-FVM is more accurate than the implicit methods at all spatial resolutions although the implicit FDM (CFL<1) approaches the high accuracy of the TVD-FVM. Only at very high resolutions (6.25 m), the implicit FDM method is approaching the accuracy obtained with the TVD-FVM. At lower spatial resolutions (>10 m) the numerical accuracy of the TVD-FVM is significantly higher compared to the accuracy obtained with the implicit methods at the cost of only-a 1050 slightly increased, without requiring additional computation time. that we optimized due by a to the vectorized implementation of the TVD-FVM. To achieve the same numerical accuracy as the TVD-FVM at 500 m spatial resolution (RMSE = 18.17, model runtime = 2.89 seconds), the implicit method (CFL<1) would need to be evaluated at 150 m which would take 12 times longer (model runtime = 36 sec) (Fig. 5). From Fig. 5 it can be derived that it would for example take for 12 times longer to obtain the accuracy of the river processes obtained with a TVD-FVM at 500 m (RMSE = 18.17, model runtime = 2.89 seconds) with an implicit method (cfl<1, at 150 m, model runtime = 36 sec).

**4.1.3. River incision and catchment wide erosion rates**

We hypothesize that apart from river profile evolution, the diffusive nature of commonly applied FDM is not restricted to the simulation of river longitudinal profiles but has systematic consequences for accurate simulation of river knickpoints will 1060 influence-other measures derived from simulations with LEMsslandscape evolution as a whole. Such measures include catchment-wide erosion rates that-often constitute the basis for model-field data comparison and model parametrization (Gasparini and Whipple, 2014; Moon et al., 2015) REFS). In order to investigate the sensitivity of LEM-derived catchment wide erosion rates to different numerical schemes of the river incision model, we first create use the -a-steady-state artificial landscape that we initialize with uniformly distributed random elevation values between 0 and 50 m on a 50 km × 100 km grid with a spatial resolution of 100 m (Movie S3)described in the previous experiments (section 4.1.2). Landscape evolution is 1065 simulated using Dirichlet boundary conditions and by inserting spatially and temporally uniform vertical uplift of 1 km Myr4 over a period of 150 Myr. Outer model timesteps are set to 5 × 104 yr. Parameter values for river incision and hillslope response are constant in space and time and are reported in Table 1. Figure 5 shows the resulting steady state landscape. Similar to these experiments simulations outlined in section 4.1.2, we imposed four consecutive uplift pulses of equal magnitude to this 1070 artificial landscape (Fig. 5). but here, Uuplift pulses have a wavelength of 1.25 Myr-and an amplitude of  $1.5 \times 10^3$  m yr-1 (Fig.  $\frac{6}{2}$  and TTLEM is run over 5 Myr-with main model time steps of  $5 \times 10^{4}$  yr, again with Dirichlet boundary conditions and a plaplanform fixed drainage network. We use two spatial resolutions (100 m and 500 m) and three different numerical methods (implicit FDM without time steptime step limitation, implicit FDM with time steptime step limitation (CFL condition applied) and TVD-FVM) to simulate river incision. When applicable, tThe maximum length of the inner time steptime step is set to 3

| Formatted: Font color: Text 1                                                        |
|--------------------------------------------------------------------------------------|
| Formatted: Font color: Text 1                                                        |
| Commented [gg1]: Correct ? Counteracts ? Trade-off ?                                 |
| Formatted: Font color: Text 1                                                        |
| Commented [WS2]: I think you can safely delete that part
It is repetitive. |
| Commented [WS3]: This part can be deleted, too.                                      |

**Formatted: Heading 1**

[revised manuscript text omitted]
., 2015) that can be assessed using . LEMs. The dynamics of drainage networks and divides (Willett et al., 2014) and the nonlinear models involved, however, entail that LEMs can hardly rely on analytical solutions. (Fox et al., 2014), but require numerical solvers of the governing PDEs. The successful use of these simulation tools thus requires knowledge about their numerical accuracy with high numerical accuracy are thus needed to capture transiency correctly, yet . Despite the growing interest in the development and use of LEMs, the assessment of LEM numerical accuracy has fallen short, yet. We show that most commonly applied first order accurate numerical methods introduce numerical diffusion and smear discontinuities that are inherent in Formatted: No Spacing

(23)

- transient landscapes. To overcome this problem, we present a higher order flux limiting scheme referred to as TVD FVM. We exemplify the use of this technique by simulating the upward migration of Knickpoints knickpoints and the evolution of river longitudinal profiles as well as horizontal tectonic movements in river systems are of particular concern to geomorphologists as their analysis reveals insights into the tectonic and climatic controls on evolving landscapes. However, no analytical solution exists that allows to simulate river incision for changing drainage areas (Fox et al., 2014). Because drainage networks and drainage divides evolve in dynamic ways (Willett et al., 2014), the analysis of transient landscapes must thus rely on numerical methods, although analytical models can be applied in specific cases (Perron and Royden, 2013). Similarly, current grid-based models do not allow to accurately simulated the evolution of a landscape subject to tectonic shortening with a spatially variable velocity field. We present a higher order flux limiting scheme (referred to as TVD FVM) that overcomes this problem of numerical diffusion...
- 1160 Our analysis of numerical solvers focusfocusseed-s on three interrelated numerical-issues: numerical accuracy, spatial resolution and computational efficiency. Adopting highly simplifying assumptions allowed us to benchmark the solvers against analytical solutions. Our focus wasis on testing an implicit FDM against TVD-FVM. The implicit FDM has numerousseveral desirable properties-advantages. It is unconditionally stable and tolerates time step lengths exceeding those prescribed by the CFL criterion. LEMs are often run over time spans of millions of years and the CFL criterion is dictated by a few DEM-grid cells with high upslope areas. Thus, Adopting an implicit scheme is therefore potentially interesting tempting as it allows to significantly decreaseing the computational time while it enablesing simulations at high spatial resolutions. Our results, however, show that this major advantage vanishes if the aim of an LEM simulation is to capture transiency correctly. For CFL ≥ 1, the implicit FDM introduces significant numerical smearing, and for CFL >> 1, the approach tends to insert artificial shockwaves of uplift as fault movements are modelledbecause gradual uplift is approximated by a step function if time steps are (very) large, as stepwise functions rather than continuously.
- For time step lengths approaching those prescribed by the CFL criterion, we show that computational gains by implicit FDM 1175 are marginal compared to TVD-FVM. The TVD-FVM code can be vectorized, i.e. it exploits single-instruction multiple-data parallelism to save CPU time. The is performance gain may not be reached by the implicit FDM requires a lower despite the lower number of numerical operations required, as this method must sequentially but loop through all stream network nodes need to be treated sequentially. However, we have not fully exploited ways to improve the computational performance of the implicit FDM such as processing individual drainage basins in parallel (Braun and Willett 2013). While unexplored in our 1180 study, we expect that separating the data by drainage basins will likely add significant computation and communication overhead.-Simulations at higher spatial resolutions increase the numerical accuracy and may balance the low accuracy of the implicit FDM. Our results indicate that there is indeed a strong gain in numerical accuracy for all methods (Fig. 4 and 5) with increasing spatial resolution. However, to achieve the same numerical accuracy as the TVD-FVM, the implicit method with a CFL<1 constraint would-requires the use of spatial resolution that is ca. 3three\* times higher, resulting in a computation time 1185 that is ca. spatial resolutions and 12 twelve times the CPU time higher (Fig. 5). In summary, while a first order implicit scheme is stable and accurate for long-term, steady-state solutions (Braun and Willett 2013), it has severe shortcomings whenin accurately simulating transient landscape evolution caused by knickpoint propagation in detachment limited erosional basins. These shortcomings can, to a large extent, be avoided by using a TVD-FVM, a finding that can also be transferred to the nonlinear river incision model (n≠1) (Campforts and Govers, 2015). (Campforts and Govers (2015)).

for such simulations.

considering only linear river incision (n=1), spatially and temporally constant parameter values, uplift and precipitation. TTLEM supports temporal and spatially variable input values for all these parameter <del>a the</del> 1195 erodibility weighting matrix (wx) or contributing drainage area weighting matrix (wx). The impact of non-linear river incision is discussed in detail in Campforts and Govers (2015). Currently TTLEM does not yet support transport limited river fluvial processes, peither glacial erosion or a bedrock/regolith interface to simulate soil evolution processes (Campforts et al. 2016). TTLEM uses D8 routing to update the drainage network during model simulations. Dinf (or Doc) is the flow routing scheme hillslopes (Pelletier, 2008), Howe sent flow 1200 domain of the landscape and thus the flow routing scheme on hillslopes is of minor significance. Neverthele channelized domain Dinf has advantages over D8 since it enables diverging flows braidplains. The current implementation of TTLEM, however, focuses on the modelling of detachment-limited systems or bedrock rivers where divergent flows are usually confined by valley walls. This is also consistent with other models such as ape (Braun and Willett, 2013) and DAC (Goren et al., 2014) models that use the D8 flow routing scheme. Nonetheless, 1205 we do not exclude to implement Dinf or other multiple flow direction algorithms in a future version of TTLEM, in particular the topological sorting algorithm (Braun and Willett computation of flows on thus derived networks.

we focus on the numerical accuracy of landscape evolution models, we focused on relatively simple simulations

Field Code Changed

What are the implications of numerical diffusion of transient river profiles for LEMs in general? A performance analysis 1210 allowed to evaluate the computational efficiency and the numerical efficiency of the different schemes implemented in TTLEM. In order to perform this analysis we implemented an analytical slope patch method for the stream power law being resolution independent. The analytical solution functions as a robust benchmark to evaluate not only the numerical accuracy of the river incision methods but also offers a tool to evaluate model performance in general. The performance analysis demonstrates (i) that the numerical methods (the implicit FDM method and the TVD-FVM) converge at high resolutions. 1215 Moreover, the analysis shows how the implicit method is only marginally performing better in terms of computational timesperformance for similar resolutions which is due to the fact that implicit schemes cannot be vectorized (see section 4) and river cells only occupy part of the landscape. The performance analysis shows how implicit methods without a restriction on the time step does not converge, partly due to the increased amount of numerical smearing introduced in the solution for CFL>1 and partly due to the fact that uplift in inserted to abruptly in the model if CFL>>1. Therefore, the main advantage of 1220 an implicit scheme, i.e. being unconditionally stable against varying time steps vanishes as also implicit schemes require the definition of an inner time step in order to properly simulate river incision. In Fig. 4, two extremes are shown, i.e. a configuration where CFL<1 and one where CFL>>1. One could argue that intermediate solutions (e.g with CFL closer to 1) would result in more desirable results than the ones shown Fig. 4. This is true but, given that computational gains are marginal and numerical accuracy will never be higher than the implicit method simulated at CFL<1 (solid blue lines), we see little 1225 reason to follow such an approach when simulating transient landscape evolution. In summary, we conclude that a first order implicit scheme is not suited to properly simulate propagating knickpoints in detachment limited erosional basins. First order implicit methods are therefore only suited to simulate configurations where transiency, caused by local base level falls, tectonic faults or lithological contacts can be considered to be minor.

Our simulations show that optimizing numerical schemes of LEMs is far from being only a numerical exercise. The We also show that the impact of the numerical scheme used to simulate detachment limited river incision on model outcomes is substantial and is not limited to river profile development alone. Hillslopes adjust to local base\_level changes dictated by river incision. Hillslope denudation rates therefore must thus — at least partly — reflect the geometry and dynamics of a knickpoint and will respond differently to whether it is a diffuse signal that is the result of relatively slow, continuous uplift on the one hand and or a sharp discontinuity migrating upstream caused by a rapid base level drop of major fault activity on the other hand. Our simulations show that, depending on the spatial and temporal resolution, catchment wide erosion rates are more responsive to uplift when fluvial incision is calculated by derived from the TVD-FVM rather than byin comparison to the implicit -FDMs. This is because Ffirst order (explicit and implicit) FDMs fail to properly reproduce transient incision waves

(Campforts and Govers, 2015) due to knickpoint smoothing. This also with the effect that the smoothing propagates to inferred
 rates of affects hillslope denudation as the drop in hillslope base level due to the passage of a knickpoint is smeared out in time when smoothing occurs. The response of -and that Ccatchment wide erosion rates to uplift will therefore also be smoothed, resulting in significantly lower peak erosion rates. are smeared over geological time. Our results show that this effect will not be strong in catchments in direct vicinity to faults, but is. This effect will be most significant in upstream catchments which are far away from the base level as smoothing increases with time and knickpoint migration distance.s further upstream. Empirical studies that aim to link their findings from e.g. detrital cosmogenie nuclide-derived denudation rates to LEMs may consider that potential bias introduced by commonly used FDMs. Thus, the use of a shock preserving method such as TVD FVM is strongly recommended for accurate simulations of transient landscapes.

**1250**

[revised manuscript text omitted]

1320 advected with the tectonically imposed velocity field (e.g. Herman and Braun, 2006). In TTLEM We implementedHere, the TVD-FVM solvers are implemented in the simulation toolin TTLEM that performs all calculations onusing a fixed grid, gridded datasetswhich has some advantages., avoids these techniques but rather attempts to run on rectangular grids with a maximum of accuracy. We chose so for several the following reasons,: First, input data such as topography, climate, lithology or tectonic displacement fields are typically available as raster datasets and thus require only minor modifications before they

| Field Code Change | d |  |
|-------------------|---|--|
|                   |   |  |
| Field Code Change | d |  |
| Field Code Change | d |  |
|                   |   |  |
| Field Code Change | d |  |
| Field Code Change | d |  |
|                   |   |  |

|     | Field Code Changed |
|-----|--------------------|
|     |                    |
| -(1 | Field Code Changed |
|     |                    |

|   | Formatted: Font: Not Italic, Font color: Text 1 |
|---|-------------------------------------------------|
| 1 | Formatted: Font: Not Italic, Font color: Text 1 |

Field Code Changed

- 1325 can be used whereas irregular grids require substantial preprocessing. Second, TTLEM output can instantly be analyzed and visualized using the TopoToolbox library (Schwanghart and Kuhn, 2010; Schwanghart and Scherler, 2014) or any other geographic information system. Thus, while irregular grid geometries and flexible grids may have some advantages over rectangular grids-with respect to numerical accuracy, TTLEM's implementation of numerically highly accurate algorithms strongly reduces reduce the shortcomings of rectangular grids while facilitating straightforward processing of model in- and 1330 output. therefore enhancing the ease of modelling.
- As we focus on the numerical accuracy of landscape evolution models, we focused on relatively simple simulations considering only linear river incision (n=1), spatially and temporally constant parameter values, uplift and precipitation. Nonetheless, TTLEM supports temporal and spatially variable input values for all these parameters, e.g. by changing the erodibility weighting matrix (wk) or contributing drainage area weighting matrix (wk). The impact of non-linear river incision is discussed 1335 in detail in. Currently TTLEM does not yet support transport limited river fluvial processes, neither glacial erosion or a bedrock/regolith interface to simulate soil evolution processes (Campforts et al., 2016). TTLEM uses D8 routing to update the drainage network during model simulations. Dinf (or  $D\infty$ ) is the flow routing scheme of choice to represent flow on hillslopes (Pelletier, 2008). However, in TTLEM fluvial erosion is limited to the channelized domain of the landscape and thus the flow routing scheme on hillslopes is of minor significance. Nevertheless, even in the channelized domain Dinf has advantages over 1340 D8 since it enables diverging flows on landforms such as alluvial fans and braidplains. The current implementation of TTLEM, however, focuses on the modelling of detachment-limited systems or bedrock rivers where divergent flows are usually confined by valley walls. This is also consistent with other models such as Fastscape (Braun and Willett, 2013) and DAC (Goren et al., 2014) models that use the D8 flow routing scheme. Nonetheless, we do not exclude to implement Dinf or other multiple flow direction algorithms in a future version of TTLEM, in particular since the topological sorting algorithm (Braun and Willett, 1345 2013; Heckmann et al., 2015) is equally suitable for the efficient computation of flows on thus derived networks.

TTLEM offers users the flexibility to address a number of issues. It allows users to define different initial conditions such as a flat surface, a randomly disturbed surface or a DEM of a real landscape. TTLEM particularly benefits from the adoption of highly efficient drainage network algorithms that outscore GIS implementations in terms of computational efficiency while 1350 maintaining their ability to handle the artefacts (artificial topographic sinks) pertinent in real world DEMs (see Table 1 in Schwanghart and Scherler (Schwanghart and Scherler, 2014)). TTLEM provides access to different models of hillslope denudation, and allows to model tectonic displacement at any desirable level of detail, Finally, TTLEM provides different numerical schemes to solve the governing equations allowing users to trade off between computational efficiency and accuracy. To our knowledge, such LEM versatility is hitherto inexistent and thus adds to the plethora of available LEMs 1355 (Valters, 2016). Its ability to be directly run on available DEMs renders TTLEM a simulation environment to explore trajectories of landscape evolution under different scenarios of geomorphological, climatological and tectonic controls.

**6. Conclusion**

1360

Most eroding landscapes are in a transient state characterized by dynamic river networks -that can be assessed using LEMs. The dynamics of drainage networks and divides and the nonlinear models involved, however, entail that LEMs can hardly rely on analytical solutions alone, but requirerequiring numerical methods to solve solvers of the governing PDEs. The successful use of these simulation tools however requires knowledge about their numerical accuracy. Despite the growing interest in the development and use of LEMs, the accuracy assessment of the numerical methods used has received little attention. LEM numerical accuracy has fallen short, yet. We show that the most commonly applied first order accurate numerical methods introduce numerical diffusion and artificially smoothen ear-the discontinuities that are inherent into transient landscapes. To 1365 overcome this problem, we present a higher order Total Variation Diminishing Finite Volume Method referred to as TVD-FVM.TTLEM v1.0 is a raster based Landscape Evolution Model (LEM) contained within TopoToolbox. It allows using a flux

limiting Total Variation Diminishing Finite Volume Method (TVD FVM) to solve the stream power law and to simulate lateral displacements. The TVD-FVM solves river incision much more accurately than the traditional schemes: this does not only affect river development but also -which is reflected in-catchment wide erosion rates. The magnitude of the errors related to numerical smearing Ddepend ing on the spatial and temporal resolution used as well as on the position of the catchment in the 1370 landscape. during model runs, first order implicit methods to simulate river incision lead to catchment wide erosion rates which smeared out over the simulated time span and does not allow to properly capture transient landscapes response. The fact that the impact of numerical schemes is not only altering simulated topography but also simulated erosion records rates is of utmost importance in the light of the current debate research efforts which aim at using ere-long term erosion histories are 1375 increasingly used\_to unravel the coupling\_upliftclimate, \_\_erosion\_and\_ e-climateuplift enigma: however, such long-term simulations are not the only ones for which an accurate representation of knickpoint dynamics is necessary. dynamics. The A 2D version of the TVD-FVM, on the other hand, allows to accurately simulate the impact of lateral tectonic displacement in a fixed grid environment, which facilitates the incorporation of this process in many existing LEMs that use such a structure.

1380 The TVD\_FVMs are implemented in the open access raster based Landscape Evolution Model (TTLEM) contained within TopoToolbox and featuring TTLEM features a range of hillslope response schemes to simulate hillslope processes and allows accurate simulation of lateral tectonic displacements, for example due to tectonic shortening. The combination of geomorphological laws to capture landscape response to changes in both internal (e.g. tectonic configurations) and external (e.g. climate changes) forcings forcing provides the community with a novel tool to accurately reconstruct, predict 1385 and explore landscape evolution scenarios over different spatial and temporal timescales. In its current form, TTLEM is limited to uplifting, fluvially eroding landscapes. Further development will allow to integrate other processes (e.g. glacial erosion) as well as the explicit routing of sediment through the landscape.

**Code availability,**

1390 TTLEM 1.0 is embedded within TopoToolbox version 2.2. The source code and future updates can be downloaded from the GIT repository https://github.com/wschwanghart/topotoolbox. TTLEM is platform independent and requires MATLAB 2014b or higher and the Image Processing Toolbox. Documentation and user manuals for the most current release version of TopoToolbox and TTLEM can be found at the GIT repository in the help folders of the software. The user manual of TTLEM includes three tutorials which can be accessed from the command window in MATLAB. To get started: download and extract 1395 the main TopoToolbox folder from the repository to a location of your choice. Add the folder to the Matlab search path by entering the following code in the command window addpath(genpath('C:\path\...\TT\_folder')). The software package comes with three examples which can be initiated from the command window by entering TTLEM\_usersguide\_1\_intro; TTLEM usersguide 2 Synthetic model run or TTLEM usersguide 3 Synthetic Geological Configuration. These tutorials are also documented in the Help folder of ttlem. The source code for the solution of the one dimensional Stream Power Law 1400 (SPLM) can be downloaded from the GIT repository https://github.com/BCampforts/SPLM. SPLM contains the solution of the 1D river incision codes including four examples.

**Acknowledgements**

1405 1410

This work was motivated by the meeting "Landscape evolution modelling - bridging the gap between field evidence and numerical models" in Hannover, 21-23, October 2015, that was organized by the FACSIMILE network and funded by the Volkswagen Foundation. Additional support comes from the Belgian Science Policy Office in the framework of the Interuniversity Attraction Pole project (P7/24): SOGLO - The soil system under global change. Numerical simulations were performed in the MATLAB environment (2015b) using numerical schemes as referred to in the text. Computational resources and services used to evaluate model performance were provided by the VSC (Flemish Supercomputer Center), managed by the Research Foundation - Flanders (FWO) in partnership with the five Flemish university associations. We are grateful to the IDYST group of the University of Lausanne and in particular Frédéric Herman and Aleksandar Licul for inspiring discussions

on numerical methods and Nadja Stalder for the figure design. We further thank Taylor Perron for sharing his source code. We also thank two anonymous reviewers for constructive feedback that improved the manuscript.

[revised manuscript text omitted]

Figure 8. Spatial pattern of erosion rates auring one model timesteptime step when simulating tandscape evolution with the4
 flux limiting TVD-FVM versus the first order implicit FDM. (a) simulation at a resolution of 100 m where the timesteptime step of the implicit method is not constrained (b) simulation at a resolution of 100 m where the timesteptime step of the implicit method is constrained with the CFL criterion (c) simulation at a resolution of 500 m where the timesteptime step of the implicit method is not constrained (d) simulation at a resolution of 500 m where the timesteptime step of the implicit method is constrained (d) simulation at a resolution of 500 m where the timesteptime step of the implicit method is constrained with the CFL criterion.

**Figure 9.** Impact of numerical schemes when simulating horizontal shortening on a fixed grid. Left: extract from synthetically produced DEM from Fig. 52. Middle: horizontal shortening in two directions simulated with a 2D explicit first order Godunov Method (GM). Right: horizontal shortening in two directions simulated with a 2D explicit flux limiting TVD-FVM.

1645

**Figure 10.** (a) Amount of numerical diffusion  $(D_N)$  introduced in the system when simulating lateral tectonic displacement in two directions as a function of raster resolution. The grey zone indicates the range of naturally observed diffusion rates. (b) The ratio between the amount of numerical diffusion for the first order Godunov Method (GM) versus the flux limiting TVD-FVM.

52

**Table 1. Model parameters used for the TTLEM simulations.**

| Parameter                 | Units                                | Figure 1            | Figure
2               | Figure 4-5                                             | Figure 6-8                                             | Figure 9-10                                      | Figure 2A                                         |
|---------------------------|--------------------------------------|---------------------|---------------------------|--------------------------------------------------------|--------------------------------------------------------|--------------------------------------------------|---------------------------------------------------|
|                           |                                      |                     |                           | Initalization                                          |                                                        |                                                  |                                                   |
| InitialSurface            |                                      | flat, 1D            | random                    | synthetically
produced
DEM
shown in Fig.
2 | synthetically
produced
DEM
shown in Fig.
2 | synthetically
produced DEM
shown in Fig. 2 | Tujunga SRTM                                      |
| UpliftPattern             |                                      | no uplift           | uniform                   | uniform                                                | uniform                                                | Lateral Displacement                             | -                                                 |
| UpliftRate
SpatialStep | m yr -1
m              | 0
100            | $1 \times 10^{-3}$
100 | 0 - 3×10 -3
varying                      | 0 - 3×10 -3
100 - 500                    | 0
varying                                     | 0
30                                           |
|                           |                                      |                     | _                         |                                                        |                                                        |                                                  |                                                   |
|                           |                                      |                     | 150 ×                     | mputational para                                       |                                                        |                                                  |                                                   |
| TimeSpan                  | yr                                   | $1 \times 10^{6}$   | 10 6           | $1 \times 10^{6}$                                      | $5 	imes 10^6$                                         | $1 \times 10^{6}$                                | $5 	imes 10^5$                                    |
| TimeStep (outer)          | yr                                   | ca. $6 \times 10^3$ | $5	imes 10^4$             | $5 	imes 10^4$                                         | $5	imes 10^4$                                          | resolution dependent